# CIRQRS: Evaluating Query Relevance Score in Composed Image Retrieval

## Abstract

Composed Image Retrieval (CIR) retrieves relevant images using a reference image and accompanying text that describes how the desired images differ from the reference. However, the commonly used evaluation metric Recall@k only checks if the target image is retrieved, without considering the relevance of other images to the query, potentially leading to user dissatisfaction. We introduce Composed Image Retrieval Query Relevance Score (*CIRQRS*), an evaluation metric that scores each retrieved image based on its relevance to the query, offering a comprehensive evaluation. *CIRQRS* is trained using a reward model objective to prefer highly relevant, *positive* images over less relevant, *negative* ones. We propose a strategy motivated by self-paced learning to dynamically adjust the negative set based on the relevance of each image by using *CIRQRS*'s current training status. To validate *CIRQRS*'s ability to measure relevance, we created the human-scored FashionIQ (HS-FashionIQ) dataset and compared it with scores from human evaluators. *CIRQRS* correlates with human scores 2.625 times better than Recall@k, highlighting its superior ability to capture relevance. Additionally, by ranking images based on their *CIRQRS*, we check if the target image appears in the top k. The results show that *CIRQRS* achieves state-of-the-art performance on two representative CIR datasets, CIRR and FashionIQ.

## 1 Introduction

Recent developments in multi-modal AI (Radford et al., 2021; Li et al., 2022; 2023) have transformed image search by using text and images as inputs, moving beyond traditional text-only queries. Using a bimodal query (a reference image and relevant text), Composed Image Retrieval (CIR) (Lee et al., 2021; Bai et al., 2024; Chen et al., 2024) retrieves images from a large corpus based on user-specified modifications. Figure 1 shows an example where a user provides a shirt image with the text 'blue t-shirt with short sleeves.' The system retrieves images that reflect these queries, such as modifications in color or style. CIR enhances search precision, particularly in cases where describing visual details is challenging with text alone, making it valuable for applications in e-commerce and internet search.

Despite progress in CIR, the widely adopted evaluation metric, Recall@k, falls short of capturing user satisfaction. While user satisfaction improves with the number of relevant items retrieved (Al-Maskari & Sanderson, 2010), Recall@k only checks whether the target image is retrieved, ignoring the relevance of other retrieved images. As shown in Figure 1, Recall@k scores 0 despite retrieving relevant images or 1 despite including irrelevant images. Defining relevance-based metrics in CIR is challenging due to the complexity of attribute modifications (e.g., color and style) and the difficulty of quantifying the relevance of each retrieved image.

We propose Composed Image Retrieval Query Relevance Score (*CIRQRS*), an evaluation metric that addresses the limitation of Recall@k. *CIRQRS* assigns score to each retrieved image based on its relevance to the query. To achieve this, we follow a reward model training objective (Ouyang et al., 2022), where *CIRQRS* is trained to maximize the likelihood that a highly relevant image, *positive image*, is preferred over a less relevant image, *negative image*. CIR datasets typically consist of triplets (reference image, relative text, and target image), and we set the target image most relevant to the query as positive. However, selecting appropriate negatives is challenging, as they are not predefined and must be less relevant than the target, but not entirely irrelevant. To overcome this

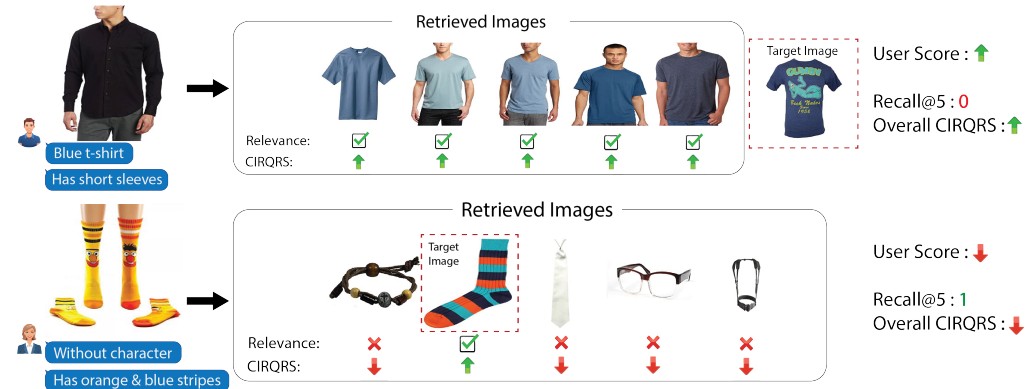

**Figure 1:** Results using Bi-BLIP4CIR (Liu et al., 2024) (top) and CLIP4CIR (Baldrati et al., 2023) (bottom) with FashionIQ dataset (Wu et al., 2021). In the top example, Recall@5 is 0 despite retrieving relevant images and in the bottom example, Recall@5 is 1, but the images are irrelevant. This misalignment highlights the problem of Recall and the advantage of our new metric, *CIRQRS*, which better aligns with human judgments and preferences.

problem, we propose a new training strategy inspired by self-paced learning, which dynamically adjusts the negative set based on the difficulty and relevance of each image according to the current training status of the model. This approach introduces increasingly difficult examples over time, aiding convergence and enhancing the performance of *CIRQRS*.

To evaluate the validity of *CIRQRS* in measuring relevance, we created the human-scored FashionIQ (HS-FashionIQ) dataset and compared *CIRQRS* with scores provided by human evaluators. Participants were shown two sets of retrieved images per query from different CIR models and rated their relevance on a 5-point Likert scale (Likert, 1932). We analyzed the correlation between *CIRQRS* and human scores, as well as Recall@k and human scores. The results show that *CIRQRS* achieves a Spearman correlation (Spearman, 1961) of 0.42 with human scores, 2.625 times higher than 0.16 in Recall@k. Additionally, when *CIRQRS* was higher on a set of retrieved images, users preferred that set 75% of the time, compared to 58% for Recall@k. This confirms *CIRQRS*'s better alignment with user preferences. The HS-FashionIQ dataset will be publicly released, with additional survey details provided in Section 4.

Additionally, we evaluate the effectiveness of *CIRQRS* using Recall@k by sorting the candidate images according to *CIRQRS* and retrieving the top-k images. If the goal of maximizing the preference for relevant images is achieved, the target image should rank higher than others in the corpus. We assess *CIRQRS*'s performance with Recall@k on two datasets: CIRR (Suhr et al., 2018) and FashionIQ (Wu et al., 2021). The results show that *CIRQRS* achieves state-of-the-art CIR performance on both datasets.

In summary, our contributions are as follows:

- We propose *CIRQRS*, an evaluation metric in CIR that overcomes the limitations of Recall@k by scoring each retrieved image's relevance to the query, providing a user-centric performance measure.

- We introduce a self-paced learning-inspired strategy that dynamically refines the negative image set during training, based on the *CIRQRS*'s current perception of image relevance. This enhances the *CIRQRS*'s ability to rank images based on the query relevance.

- We created the human-scored HS-FashionIQ dataset to evaluate the validity of *CIRQRS* by comparing it with human-provided scores. The results show that *CIRQRS* correlates 2.625 times more strongly with human ratings than Recall@k.

- We evaluate *CIRQRS*'s effectiveness using Recall@k, demonstrating state-of-the-art performance on the CIRR and FashionIQ datasets by ranking target image higher in the candidate set.

## 2 RELATED WORK

**Vision-Language Foundation Model.** Vision-Language Models (VLMs) have gained attention for their ability to integrate multimodal data. Transformer-based architectures effectively handle both visual and language inputs (Li et al., 2019; Lu et al., 2019). Contrastive learning methods, which align visual and language modalities, have significantly improved performance in VLMs (Jia et al., 2021; Radford et al., 2021). Research has also explored architectures that combine features from both modalities. For instance, Flamingo (Alayrac et al., 2022) and BLIP (Li et al., 2022) use cross-attention, where visual hidden states from the vision encoder are inserted into cross-attention layers within the text encoder layers. BLIP2 (Li et al., 2023) and QWEN (Bai et al., 2023) utilize pre-trained image and text encoders with learnable networks that bridge the gap between modalities. *CIRQRS* adopts BLIP2, using its image and text encoders with the Q-former module to handle modality gaps. We chose BLIP2 for its efficient combination of image and text processing, requiring minimal training of the Q-former module.

**Composed Image Retrieval.** The CIR task retrieves images using multimodal input features. A common approach is feature fusion, where the reference image and text are jointly embedded and compared against embeddings of candidate images (Vo et al., 2019; Dodds et al., 2020; Liu et al., 2021; Baldrati et al., 2023). Bi-BLIP4CIR (Liu et al., 2024) trains the text encoder using bidirectional training to capture both text directions of a given relation. CASE (Levy et al., 2024) leverages BLIP (Li et al., 2022) cross-attention architecture to perform an early fusion between the modalities. Other approaches transform images into pseudo-word embeddings or sentence-level prompts for text-to-image retrieval (Liu et al., 2023; Saito et al., 2023; Bai et al., 2024). MGUR (Chen et al., 2024) introduces an uncertainty loss for coarse-grained retrieval, and SPN4CIR (Feng et al., 2024) proposes a data generation method to scale positive and negative samples using multimodal LLMs. However, all previous works are evaluated using the metric Recall@K, which has inherent limitations in evaluating retrieved image sets. *CIRQRS* addresses this issue by proposing a new evaluation metric that captures the overall relevancy of the retrieved set rather than relying solely on target images as an anchor.

**Self-Paced Learning.** Curriculum learning trains models with progressively harder samples to improve the model performance (Bengio et al., 2009). Self-paced learning extends this by dynamically determining the difficulty of each sample during training based on the model perception (Kumar et al., 2010). Further research explores application across tasks (Lee & Grauman, 2011; Tang et al., 2012) with various criteria used to rank samples, such as objectness function (Jiang et al., 2014) or prior knowledge (Jiang et al., 2015). To train *CIRQRS* as an accurate scoring model, it is crucial to select an appropriate negative image that is less relevant than the target image. We use a self-paced learning-inspired strategy that dynamically adjusts the negative set based on the difficulty and relevance of each image according to the *CIRQRS*'s training progress.

## 3 METHODOLOGY

### 3.1 OVERVIEW

We denote a CIR dataset as $\mathbb{D} = \{d_i \mid i = 1, \ldots, N_d\}$, where each data point consists of a reference image, relative text, and a target image, i.e., $d_i = \{x_{I_i}, x_{T_i}, y_{I_i}\}$. The goal of CIR is to retrieve a set of images from the entire candidate image corpus $\mathbb{I} = \{I_j \mid j = 1, \ldots, N_{img}\}$, including the target image $y_I$, where the retrieved images reflect the specified relative text $x_T$ while preserving the visual properties of the reference image $x_I$. However, the current evaluation metric in CIR, Recall@k, only checks if $y_I$ is among the top-k retrieved images without considering the relevance of other retrieved images to the query, potentially causing user dissatisfaction (Al-Maskari & Sanderson, 2010). We propose *CIRQRS*, a new evaluation metric that scores each retrieved image based on its relevance to the query. As shown in Figure 2, *CIRQRS* is trained to maximize the probability that the highly relevant image is preferred over the less relevant, negative image. To select appropriate negatives, we employ self-paced learning, defining the negative set as the highest *CIRQRS* images below the target and progressively reducing the negative set size to focus on increasingly difficult negatives as training progresses. The training algorithm for *CIRQRS* is provided in Appendix A.

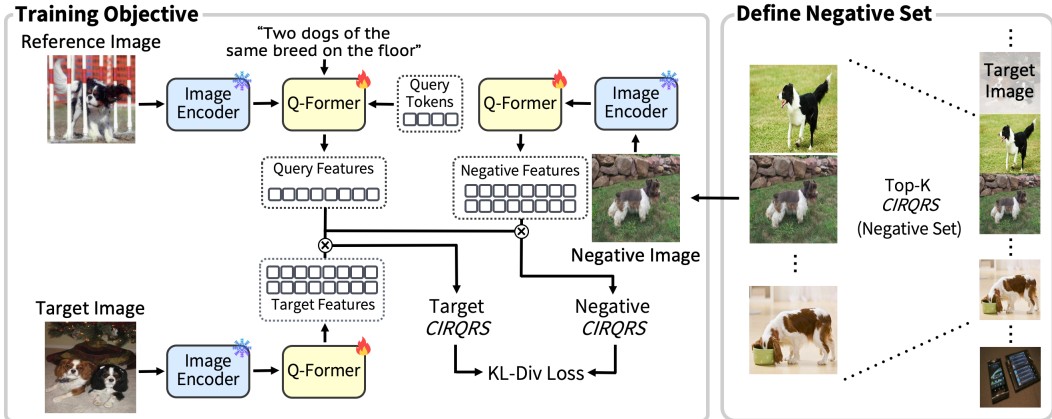

**Figure 2:** Overview of *CIRQRS*. During training, a negative image set is defined by scoring each candidate image using the current *CIRQRS* and taking the top k images with *CIRQRS* lower than the target image. We calculate the *CIRQRS* of the target image along with one random negative image and use a KL-div loss to maximize the score of the positive (highly relevant) image and minimize the score of the negative (less relevant) image.

## 3.2 SCORING MODEL

Building on recent trends in reinforcement learning with human feedback (Ouyang et al., 2022), which uses a reward model to guide generative model training, we develop a scoring model that emulates the objective of a reward model. For each candidate image $I_j \in \mathbb{I}$, we define the relevance score to the query as the inner product of the query and image embeddings:

$$s(x_{I_i}, x_{T_i}, I_j) = \frac{Q(E_{img}(x_{I_i}), x_{T_i}) \cdot Q(E_{img}(I_j))}{\tau}. \tag{1}$$

Here, $E_{img}$ is the BLIP-2 image encoder, $Q$ denotes the Q-Former, and $\tau$ is the learned BLIP-2 temperature. The reference image embedding serves as input to the Q-Former and the text, aligning image and text modalities via cross-attention.

To train *CIRQRS* to assign higher scores to highly relevant images over less relevant ones, we model the latent preference distribution $p^*$ using the Bradley-Terry model (Bradley & Terry, 1952), where $I_p$ represents the *positive* (highly relevant) image and $I_n$ the *negative* (less relevant) image:

$$p^*(I_p \succ I_n \mid x_I, x_T) = \frac{exp(s(x_I, x_T, I_p))}{exp(s(x_I, x_T, I_p)) + exp(s(x_I, x_T, I_n))} \tag{2}$$
$$= \sigma(s(x_I, x_T, I_p) - s(x_I, x_T, I_n)).$$

Given the triplet dataset, we set the target image most relevant to the query as the positive $I_p = y_I$. To include the negative image $I_n$, we define the dataset $\mathbb{D}^* = \{(x_I, x_T, I_p, I_n) \mid (x_I, x_T, I_p) \in \mathbb{D}, \ I_n \in \mathbb{I} \setminus I_p\}$. The model is trained by optimizing the negative log-likelihood (NLL) loss to ensure query-relevant positive images score higher than negatives:

$$\mathcal{L}_{match} = -\mathbb{E}_{(x_I, x_T, I_p, I_n) \sim \mathbb{D}^*}[\log(\sigma(s(x_I, x_T, I_p) - s(x_I, x_T, I_n)))]. \tag{3}$$

Minimizing this NLL loss is the same as minimizing the KL divergence between $p^*$ and a target distribution $p = [1, 0]$ that indicates the positive image should always score higher than the negative.

## 3.3 NEGATIVE SET DEFINITION

Due to the large size of $\mathbb{D}^*$, training on the full dataset is infeasible. To resolve this, we approximate $\mathcal{L}_{match}$ for each epoch by randomly selecting one image from a pool of negative images for each query as the negative image $I_n$. However, setting the negative pool as $\mathbb{I} \setminus \{I_p\}$ and optimizing the approximated $L_{match}$ is likely to be sub-optimal, as it may continuously select easy negative with a low *CIRQRS*. To address this, we build a smaller and more appropriate negative set for each query. For the $i$-th data point $\boldsymbol{d}_i = (x_{I_i}, x_{T_i}, y_{I_i}) \in \mathbb{D}$, the set of excluded images is defined as $\mathbb{I}_{easy_i} = \{I \mid I \in \mathbb{I} \setminus \{y_{I_i}\}, \ s(x_{I_i}, x_{T_i}, y_{I_i}) \gg s(x_{I_i}, x_{T_i}, I)\}$. Notably,

$$\forall I \in \mathbb{I}_{easy_i}, \ -\log(\sigma(s(x_{I_i}, x_{T_i}, y_{I_i}) - s(x_{I_i}, x_{T_i}, I))) \approx 0, \tag{4}$$

and the contribution of $\mathbb{I}_{easy}$ to $\mathcal{L}_{match}$ is close to zero. Therefore, we approximate the original loss (3) by focusing on the images in $\mathbb{I} \setminus \mathbb{I}_{easy_i}$ for each query, which the model considers 'hard'. Thus, we define the hard-dataset $\mathbb{D}^*_{hard_i} = \{(x_{I_i}, x_{T_i}, I_{p_i}, I_{n_i}) \mid (x_{I_i}, x_{T_i}, I_{p_i}) = \boldsymbol{d}_i \in \mathbb{D}, I_{n_i} \in \mathbb{I} \setminus \mathbb{I}_{easy_i}\}$ as our new negative set. Assuming further training keeps the model in parameter space where the loss for $\mathbb{D}^* \setminus \mathbb{D}^*_{hard}$ remains small, we reframe the original objective by minimizing the following loss function:

$$\mathcal{L}'_{match} = \mathbb{E}_{(x_I, x_T, I_p, I_n) \sim \mathbb{D}^*_{hard}}[-\log(\sigma(s(x_I, x_T, I_p) - s(x_I, x_T, I_n)))]. \qquad (5)$$

Defining the boundary and obtaining $\mathbb{I} \setminus \mathbb{I}_{easy_i}$ is challenging. The most straightforward approach would be to select $n_{neg}$ number of images with the highest *CIRQRS* for each query. However, this increases the chance of false negatives (relevant but non-target images) being included in the negative set, which disrupts model training (quantitative and qualitative evidences in Appendix B). To address this, we define the negative set by selecting $n_{neg}$ images with the highest *CIRQRS that are lower than the target image* for each query, as shown below.

$$\mathbb{I} \setminus \mathbb{I}_{easy_i} = \text{Top-}n_{neg} \left( \{I \mid I \in \mathbb{I} \setminus \{y_{I_i}\}, \; s(x_{I_i}, x_{T_i}, y_{I_i}) > s(x_{I_i}, x_{T_i}, I)\} \right) \qquad (6)$$

Our novel approach to defining hard negatives stems from our training objective. By designating the target image as the positive and a random image from the negative set as the negative, the model's objective is to increase the target image score while lowering the negative image score. Our approach excludes images with higher scores than the target from the negative set, thus relevant images are likely to retain high scores. This approach prevents well-matched images from being treated as negatives, allowing relevant images to achieve higher scores even if they are not labeled as targets.

To further aid convergence, we employ a curriculum learning strategy that gradually increases data difficulty. In the initial epochs, we define the negative set $\mathbb{D}^*_{hard}$ as $\mathbb{I} \setminus \{I_p\}$, randomly selecting negatives from the entire candidate corpus. At regular intervals, we update the negative set using the learned *CIRQRS* to form new $\mathbb{D}^*_{hard}$. The size of $\mathbb{D}^*_{hard}$ is progressively reduced, exposing the model to increasingly challenging examples, as larger $\mathbb{D}^*_{hard}$ will include easier negatives.

# 4 HS-FASHIONIQ DATASET

Evaluating the effectiveness of a new metric such as *CIRQRS* is challenging, especially in assessing how well the metric reflects the relevance of retrieved images to the given query. This highlights the necessity of creating a human-scored dataset in CIR for accurate assessment. Hence, we conducted a user survey with 61 participants to create the Human Scored-FashionIQ (HS-FashionIQ) dataset. We selected the FashionIQ dataset due to its high relevance and applicability to a broad audience, mirroring the search functionalities of e-commerce platforms.

**Data Collection Method.** Each question in the user survey consisted of two sets of retrieved images, each consisting of the top 5 results from different CIR models. For every question, two CIR models were randomly selected from the following four: CLIP4CIR (Baldrati et al., 2023), Bi-BLIP4CIR (Liu et al., 2024), CoVR-BLIP (Ventura et al., 2024), and SPRC (Bai et al., 2024). We provided queries and retrieved images from the 'shirts' or 'top tees' categories of the FashionIQ dataset to participants. Each participant was given 50 questions with a total of 100 sets of retrieved images, covering 3,050 queries in the FashionIQ validation set.

**Annotation Methodology.** For each question in the survey, participants rated each set of retrieved images on a 5-point Likert scale (Likert, 1932), where a score of 5 indicates a strong match with the query. This allows us to analyze the correlation between human scores and metrics such as recall and *CIRQRS*. Additionally, participants chose the preferred set between the two sets provided, further assessing the alignment of recall and *CIRQRS* with human preferences. To our knowledge, this is the first CIR dataset with human-scored retrieved images. An example of data from HS-FIQ is shown in Figure 5, with a detailed explanation of the data collection process in Appendix C.

**Table 1:** HS-FashionIQ Dataset: Each query has two sets of retrieved images from different CIR models, scored based on their relevance to the query.

| #Total Queries | #Shirts Queries | #Toptee Queries | #Invalid Queries |
|---|---|---|---|
| 3,050 | 1,800 | 1,250 | 307 |

**Modality Redundancy Check.** While CIR should consider both the reference image and relative text, some examples focus solely on either the image or the text. CASE (Levy et al., 2024) highlighted modality redundancy in FashionIQ, indicating that text can sometimes be more influential than the image. We instructed participants to consider both input modalities equally. We asked them to flag instances where one modality seemed irrelevant to the retrieved images, to exclude data unclear for human evaluation. A total of 307 queries are treated as irrelevant and excluded, leaving us with 2,743 valid queries in the HS-FashionIQ Dataset.

## 5 EXPERIMENTS

### 5.1 EXPERIMENTAL SETUP

**Datasets.** We evaluate the validity of *CIRQRS* on the HS-FashionIQ dataset to assess how well it scores retrieved images based on their relevance to the query, measuring correlation with human scores and alignment with human preferences. Additionally, we evaluate *CIRQRS* with Recall@k by selecting the top-k *CIRQRS* images to check if the target is included, as higher scores for relevant images should result in a higher ranking among candidates. For Recall@k evaluation, two benchmarks are used: FashionIQ (Wu et al., 2021) and CIRR (Suhr et al., 2018).

**Implementation.** We used BLIP-2 (Li et al., 2023) with a ViT-G image encoder. Following previous work (Baldrati et al., 2023), we resized images to 224×224 with a 1.5 padding ratio. *CIRQRS* is trained with AdamW optimizer (Loshchilov, 2017) for 50 epochs on CIRR and 30 epochs on FashionIQ. We defined the negative set $n_{def}$ times, warming up *CIRQRS* with the entire candidate corpus as negatives for the first $\lfloor n_{epoch}/n_{def} \rfloor$ epochs. After the warmup, we initially defined the negative set with a size of $n_{neg}$, then redefined it every $\lfloor n_{epoch}/n_{def} \rfloor$ epochs, halving its size each time. We set $n_{def}$ to 5 and 6 for FashionIQ and CIRR, respectively. We conducted our experiments using a single Nvidia RTX 3090 GPU.

### 5.2 EVALUATION WITH HS-FASHIONIQ DATASET

**Correlation with Human Score.** We evaluate *CIRQRS* correlation to human scores. Given a query $\{x_{I_i}, x_{T_i}\}$, each participant received two different sets of retrieved images, $\mathbb{S}_{1_i}$ and $\mathbb{S}_{2_i}$, and 1rated them based on their relevance to the query. To calculate the overall *CIRQRS* of a set, we averaged the *CIRQRS* of the five retrieved images within the set as the overall score. The following equation calculates for the $i$-th query and the $j$-th retrieved set:

$$CIRQRS(\mathbb{S}_{j_i}) = \frac{1}{5} \sum_{I \in \mathbb{S}_{j_i}} s(x_{I_i}, x_{T_i}, I). \tag{7}$$

With 2,743 valid queries, we obtained 5,486 sets with Recall@5, *CIRQRS*, and human scores. Since all metrics deviate from normality based on the Shapiro-Wilk test (Shapiro & Wilk, 1965) (p < .05), we used the Spearman correlation (Spearman, 1961), which is suitable for non-parametric comparison. Table 2 presents the correlation results for Recall@5 and *CIRQRS* with human scores, where a higher statistic indicates stronger alignment with human judgments, and a p-value[1] below .05 indicates statistical significance.

**Table 2:** Spearman correlation of human score with Recall@5 and *CIRQRS*.

| Metric | Statistic | P-value |
|--------|-----------|---------|
| Recall@5 | 0.16 | $p < .001^{***}$ |
| *CIRQRS* | 0.42 | $p < .001^{***}$ |

**Table 3:** Preference rate of Recall and *CIRQRS*.

| Metric | Preference Rate |
|--------|-----------------|
| Recall@5 | 0.58 |
| *CIRQRS* | 0.75 |

Table 2 shows that both *CIRQRS* and Recall@5 are statistically significance based on their p-values. However, *CIRQRS* demonstrates a stronger correlation with human scores, with a correlation value of 0.42, compared to Recall@5, which has a weaker correlation of 0.16. Since human scores are based on the relevance of the retrieved image to the query, this substantial difference suggests that

---

[1] *** in Table 2 indicates $p \leq 0.001$

**Table 4:** Performance comparison on the FashionIQ validation dataset across different methods, The best results are highlighted in bold, and the second-best are underlined.

| Method | Dress | | Shirt | | Toptee | | Average | | |
|---|---|---|---|---|---|---|---|---|---|
| | R@10 | R@50 | R@10 | R@50 | R@10 | R@50 | R@10 | R@50 | Avg. |
| CoSMo (Lee et al., 2021) | 25.64 | 50.30 | 24.90 | 49.18 | 29.21 | 57.46 | 26.58 | 52.31 | 39.45 |
| CASE (Levy et al., 2024) | 47.44 | 69.36 | 48.48 | 70.23 | 50.18 | 72.24 | 48.70 | 70.61 | 59.66 |
| AMC (Zhu et al., 2023) | 31.73 | 59.25 | 30.67 | 59.08 | 36.21 | 66.06 | 32.87 | 61.46 | 47.17 |
| CoVR-BLIP (Ventura et al., 2024) | 44.55 | 69.03 | 48.43 | 67.42 | 52.60 | 74.31 | 48.53 | 70.25 | 59.39 |
| CLIP4CIR (Baldrati et al., 2023) | 33.81 | 59.40 | 39.99 | 60.45 | 41.41 | 65.37 | 38.40 | 61.74 | 50.07 |
| Bi-BLIP4CIR (Liu et al., 2024) | 42.09 | 67.33 | 41.76 | 64.28 | 46.61 | 70.32 | 43.49 | 67.31 | 55.40 |
| FAME-ViL (Han et al., 2023) | 42.19 | 67.38 | 47.64 | 68.79 | 50.69 | 73.07 | 46.84 | 69.75 | 58.29 |
| TG-CIR (Wen et al., 2023) | 45.22 | 69.66 | 52.60 | 72.52 | 56.14 | 77.10 | 51.32 | 73.09 | 62.21 |
| DRA (Jiang et al., 2023) | 33.98 | 60.67 | 40.74 | 61.93 | 42.09 | 66.97 | 38.94 | 63.19 | 51.06 |
| Re-ranking (Liu et al., 2023) | 48.14 | 71.43 | 50.15 | 71.25 | 55.23 | 76.80 | 51.17 | 73.16 | 62.17 |
| CompoDiff (Gu et al., 2023) | 40.65 | 57.14 | 36.87 | 57.39 | 43.93 | 61.17 | 40.48 | 58.57 | 49.53 |
| SPRC (Bai et al., 2024) | **49.18** | **72.43** | 55.64 | 73.89 | 59.35 | 78.58 | 54.72 | **74.97** | 64.85 |
| *CIRQRS*-Model | 48.44 | 72.04 | **56.58** | **74.24** | **59.66** | **78.63** | **54.89** | **74.97** | **64.93** |

*CIRQRS* aligns more consistently with human evaluations, providing a more reliable measure of relevance than Recall@5.

**Alignment with Human Preferences.** We evaluate *CIRQRS* by comparing its alignment with human preferences by analyzing the *preference rate*, which is the conditional probability that Set 1 is preferred when either *CIRQRS* or Recall@5 for Set 1 is greater than or equal to Set 2. Specifically, we define preference rate as:

$$\mathbb{P}(Set\ 1 \succ Set\ 2 \mid f_{eval}(Set\ 1) \geq f_{eval}(Set\ 2)), \qquad (8)$$

where $f_{eval} \in \{CIRQRS, \text{Recall@5}\}$.

Table 3 shows that Set 1 is preferred 58% of the time when its Recall@5 is greater than or equal to Set 2, but is preferred 75% of the time in *CIRQRS*. This indicates *CIRQRS* aligns more closely with human preferences than Recall@5.

$$\mathbb{P}(Set\ 1 \succ Set\ 2 \mid \mathcal{C}(Set\ 1) > \mathcal{C}(Set\ 2) \wedge \mathcal{R}(Set\ 1) = \mathcal{R}(Set\ 2)), \qquad (9)$$

where $\mathcal{C}$ and $\mathcal{R}$ represent *CIRQRS* and Recall@K, respectively. The result of 0.73 indicates Set 1 is preferred 73% of the time when *CIRQRS* is higher, even when Recall@k is equal. This occurs because Recall@k only checks whether the target image is present in the set, without considering how relevant images are. As a result, when both sets either include or exclude the target image, Recall@k fails to capture any differences in the quality of relevance of the other images. In contrast, *CIRQRS* takes these factors into account, providing a more detailed assessment that better aligns with human preferences.

## 5.3 COMPARISON WITH STATE-OF-THE-ART CIR MODELS

We evaluate *CIRQRS* using Recall@k. Although *CIRQRS* is not designed as a CIR model, if it accurately scores images based on query relevance, the target image should rank high among the candidates. To test this, we sort all candidate images based on their *CIRQRS* and check whether the target image appears in the top k. This evaluation is performed on two benchmarks: FashionIQ and CIRR. We denote our approach as *CIRQRS*-Model, as it was originally designed as a metric rather than a model, to avoid confusion.

Table 4 summarizes the performance of various methods on the FashionIQ dataset. *CIRQRS*-Model consistently ranks first or second across all categories and achieves the best overall average. *CIRQRS*-Model achieves an improvement in Recall@10, gaining 0.96% on 'shirt' and 0.17% on average Recall@10 over the second best method, indicating the advantage of its training objective to highly score and rank the preferred target image and relevant images. Previous work (Levy et al., 2024) highlights the issue that the FashionIQ dataset has high modality redundancy, where text is

**Table 5:** Performance comparison on the CIRR test dataset across different methods, where Recall$_s$@K represents Recallsubset@K. The best results are highlighted in bold, and the second-best are underlined.

| Method | Recall@K | | | | Recall$_s$@K | | | Average | |
|---|---|---|---|---|---|---|---|---|---|
| | K=1 | K=5 | K=10 | K=50 | K=1 | K=2 | K=3 | R@5 + R$_s$@1 | Overall |
| CLIP4CIR (Baldrati et al., 2023) | 38.53 | 69.98 | 81.86 | 95.93 | 68.19 | 85.64 | 94.17 | 69.09 | 76.33 |
| Bi-BLIP4CIR (Liu et al., 2024) | 40.15 | 73.08 | 83.88 | 96.27 | 72.10 | 88.27 | 95.93 | 72.59 | 78.53 |
| CompoDiff (Gu et al., 2023) | 22.35 | 54.36 | 73.41 | 91.77 | 35.84 | 56.11 | 76.60 | 45.10 | 58.63 |
| CASE (Levy et al., 2024) | 48.00 | 79.11 | 87.25 | 97.57 | 75.88 | 90.58 | 96.00 | 77.50 | 82.06 |
| CASE Pre-LaSCo.Ca (Levy et al., 2024) | 49.35 | 80.02 | 88.75 | 97.47 | 76.48 | 90.37 | 95.71 | 78.25 | 82.59 |
| TG-CIR (Wen et al., 2023) | 45.25 | 78.29 | 87.16 | 97.30 | 72.84 | 89.25 | 95.13 | 75.57 | 80.75 |
| DRA (Jiang et al., 2023) | 39.93 | 72.07 | 83.83 | 96.43 | 71.04 | 87.74 | 94.72 | 71.56 | 77.97 |
| CoVR-BLIP (Ventura et al., 2024) | 49.69 | 78.60 | 86.77 | 94.31 | 75.01 | 88.12 | 93.16 | 76.81 | 80.81 |
| Re-ranking (Liu et al., 2023) | 50.55 | 81.75 | 89.78 | 97.18 | 80.04 | 91.90 | 96.58 | 80.90 | 83.97 |
| SPRC (Bai et al., 2024) | 51.96 | 82.12 | 89.74 | 97.69 | **80.65** | **92.31** | **96.60** | 81.39 | 84.44 |
| *CIRQRS*-Model | **53.81** | **83.30** | **90.92** | **98.27** | 79.59 | 92.15 | 96.39 | **81.45** | **84.92** |

dominant in retrieval tasks. This advantages text-based methods such as Bi-BLIP4CIR (Liu et al., 2024) and SPRC (Bai et al., 2024). Despite this, *CIRQRS*-Model achieves state-of-the-art performance, improving the overall average recall by 9.53% and 0.08% compared to Bi-BLIP4CIR and SPRC, respectively. This demonstrates *CIRQRS*'s ability to accurately score image relevance to the query.

We also evaluate *CIRQRS*-Model on CIRR, a general domain dataset. Table 5 reports various model performances on the CIRR dataset. *CIRQRS*-Model achieves the best performance across Recall@k metrics, especially on Recall@1 and Recall@5, where it outperforms the current SoTA of SPRC (Bai et al., 2024) by 1.85% and 1.18% respectively. This shows that *CIRQRS*-Model follows its training objective well and can place the target image on a high rank even compared with traditional CIR models. Despite having a high performance on Recall$_s$@k, which retrieves from a subset containing relevant images with the target, *CIRQRS*-Model did not perform the best out of the other methods. This stems from the design of *CIRQRS*-Model, where even non-target images can score higher than the target if they match the query well. The negative set is defined as images with lower *CIRQRS* than the target, preventing relevant images from being selected as negatives and allowing them to be ranked higher than the target. Despite this, *CIRQRS*-Model achieves SoTA performance on both the Recall@5 + Recall$_s$@1 average and the overall average. These results highlight *CIRQRS*'s ability to score images accurately.

## 5.4  EVALUATION OF CIR MODELS WITH *CIRQRS*

We demonstrates the applicability of *CIRQRS* as a metric by evaluating four CIR models—CLIP4CIR, Bi-BLIP4CIR, CoVR-BLIP, and SPRC—on the FashionIQ dataset. For a given query $(x_{I_i}, x_{T_i})$, where $\mathbb{S}_i$ denotes the set of retrieved images, *CIRQRS* is computed as:

$$CIRQRS(\mathbb{S}_i) = \frac{1}{|\mathbb{S}_i|} \sum_{I \in \mathbb{S}_i} s(x_{I_i}, x_{T_i}, I). \tag{10}$$

Table 6 shows the results and highlights that *CIRQRS* decrease as the size of the retrieved image set increases. This decrease occurs because the inclusion of irrelevant images reduces the average relevance of the retrieved set. The ranking of the CIR models based on *CIRQRS* scores is SPRC > CoVR-BLIP > Bi-BLIP4CIR > CLIP4CIR, demonstrating that higher-ranked models align more closely with human preferences.

To validate these findings, human preference rates for the four CIR models were computed using the metric defined in Equation 8. Since these CIR models do not inherently assign scores to retrieved images, their cosine similarity scores were used as proxies. The computed human preference rates for the models are as follows: SPRC (0.7339), CoVR-BLIP (0.7276), Bi-BLIP4CIR (0.6700), and CLIP4CIR (0.6608). Two key conclusions arise: (1) human preference rates align

**Table 6:** Performance comparison on the FashionIQ validation dataset using metric *CIRQRS*.

| Method | Dress | | Shirt | | Toptee | | Average | | |
|---|---|---|---|---|---|---|---|---|---|
| | CIRQRS@10 | CIRQRS@50 | CIRQRS@10 | CIRQRS@50 | CIRQRS@10 | CIRQRS@50 | CIRQRS@10 | CIRQRS@50 | Avg. |
| CLIP4CIR | 67.97 | 66.35 | 68.61 | 66.51 | 67.62 | 65.88 | 68.07 | 66.25 | 67.16 |
| Bi-BLIP4CIR | 71.03 | 68.86 | 69.90 | 67.51 | 70.16 | 67.64 | 70.36 | 68.00 | 69.18 |
| CoVR-BLIP | 72.07 | 69.78 | 71.41 | 68.76 | 71.38 | 68.64 | 71.62 | 69.06 | 70.34 |
| SPRC | **72.26** | **69.99** | **71.89** | **69.27** | **71.76** | **69.07** | **71.97** | **69.44** | **70.71** |

with *CIRQRS*-based rankings, confirming that *CIR models with higher* CIRQRS *scores correspond to higher human preference rates*, and (2) *CIRQRS* achieves the strongest correlation with human preference rates (0.7524), highlighting its reliability as an evaluation metric. Based on these results, we believe that evaluating CIR models with *CIRQRS* in future research will effectively reflect human preferences, making it well-aligned with the practical applications of CIR.

## 5.5 Ablation Studies

**Effect of Negative Set Definition Strategy.** To evaluate the effectiveness of our approach to defining the negative set, we compare three different strategies. The *Baseline* method defines the negative set as the entire candidate corpus $\mathbb{I} \setminus \{I_p\}$. The *Top* method selects the top $n_{neg}$ images with the highest *CIRQRS* as the negative set, initializing $n_{neg}$ at 1000, which yields the best performance. We compared our strategy, which defines the negative set as the top $n_{neg}$ images with *CIRQRS* lower than the target.

Table 7 shows the baseline method performs reasonably well compared with other methods in Section 5.3. The *Top* approach improves performance by sampling from a predefined negative set, demonstrating the effectiveness of training with harder negatives. On average, *CIRQRS* further improves by 0.40% in CIRR and 1.54% in FashionIQ than *Top*. The correlation between human scores and preference rates is consistent across all three approaches. This improvement in Recall@k suggests that, despite similar human score correlations, *CIRQRS* ranks the target image higher than other approaches. This can be attributed to the fact that as *CIRQRS* learns, many high-scoring images tend to closely match the query, making training with the *Top* method unstable and suboptimal. Our strategy mitigates this by selecting images with lower scores than the target image, which prevents highly relevant images from being selected as negative.

Figure 3 illustrates the impact of different strategies, showing Recall@1 performance on the CIRR validation set and training loss across all epochs. As described in Section 5.1, we redefine the negative set every eight epochs for the CIRR dataset in both *Top* and *CIRQRS*. Performance improves immediately after the first hard negative set is defined, compared to *Baseline*. *CIRQRS* shows more consistent, stable improvement, particularly in early training, while the sharp rise in loss after each negative set redefinition reflects the model facing harder negatives. Despite higher loss, *CIRQRS* maintains superior performance over *Top*.

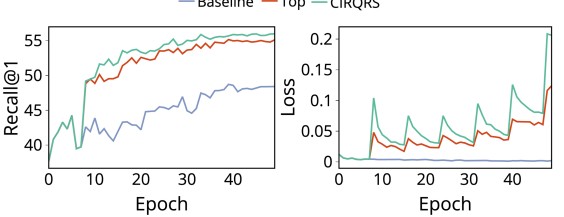

**Figure 3:** Graph of Recall@1 on the validation set and loss on the training set for the CIRR dataset.

**Figure 4:** Graph of average recall for different sizes of the initial negative pool.

**Effect on Size of Negative Set.** The difficulty of images that the model is trained on depends heavily on the size of the negative set, where a larger negative set would include easier images. In Figure 4, we compare different initial negative set sizes of $n_{neg}$ following the experiment setting in Section 5.1. We see that a lower initial $n_{neg}$ results in higher Recall@k, as it corresponds to a harder negative set and improves performance than a larger $n_{neg}$. Since we define negatives as images with

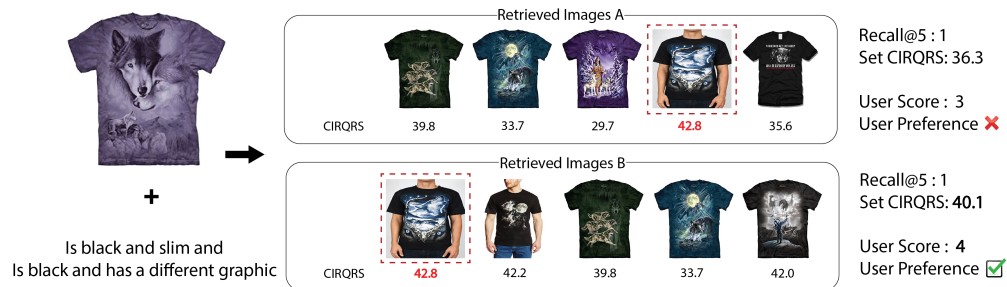

**Figure 5:** Qualitative analysis of *CIRQRS* using sample from HS-FashionIQ. Sets A and B consist of images retrieved by two different models, with the red box highlighting the target image for the given query. The human annotator preferred Set B, which aligns with the *CIRQRS*.

**Table 7:** Ablation study on defining the negative image set. (1) Baseline: The negative set is the whole candidate image corpus. (2) Top: The negative set is $n_{neg}$ images with the highest *CIRQRS* in the whole corpus for each query. **(3) Ours**: The negative set is $n_{neg}$ images with the highest *CIRQRS* in the set of images with a lower *CIRQRS* than the target image for each query.

| Method | CIRR | | FashionIQ | | |
|---|---|---|---|---|---|
| | R@5 + R$_s$@1 | Avg. | Avg R@10 | Avg R@50 | Avg. |
| Baseline | 75.96 | 81.46 | 52.22 | 73.51 | 62.86 |
| Top | 80.96 | 84.52 | 52.92 | 73.87 | 63.39 |
| *CIRQRS*-**Model** | **81.45** | **84.92** | **54.89** | **74.97** | **64.93** |

lower *CIRQRS* than the target, even a small $n_{neg}$ could properly select the set of negative images. We used an initial negative pool size of 50 for FashionIQ and 100 for CIRR.

**Qualitative Analysis.** To visualize the use of *CIRQRS* in evaluating CIR queries, we applied it to an example from the HS-FashionIQ dataset. When the target image appears in both retrieved image sets, Recall@5 of both sets is 1. However, humans can still recognize which set matches the query better, as reflected by the higher score of 4 for set B compared with 3 for set A. Using our *CIRQRS* and Equation 7, the set *CIRQRS* for set A is lower than set B's, aligning with human annotators. This difference is attributed to a purple shirt in set A, which *CIRQRS* scored lower, reducing the overall set score. Additionally, *CIRQRS* assigned a lower score to the non-black shirt than the black shirt in both sets, indicating its ability to accurately assess relevance to the query.

# 6 CONCLUSION

We introduced *CIRQRS*, a new evaluation metric that overcomes the limitation of Recall@k in CIR. Unlike Recall@k, *CIRQRS* evaluates the relevance of individual retrieved images to the query, providing a comprehensive assessment of retrieval quality. Our approach employs a reward model training objective and a self-paced learning strategy to refine the negative set, improving relevance scoring dynamically. To evaluate the effectiveness of *CIRQRS*, we created the HS-FashionIQ dataset, the first human-scored dataset in CIR. Experimental results show that *CIRQRS* achieves a significantly higher correlation with human scores than Recall@k. Using *CIRQRS* as a CIR model outperforms state-of-the-art methods across multiple CIR benchmarks, including FashionIQ and CIRR datasets.

REPRODUCIBILITY STATEMENT

We have provided a stripped down version of our experiment codes to highlight our contribution, which includes our model, training pipeline, and evaluation. We followed the dataset preprocessing procedure outlined in SPRC (Bai et al., 2024). Our model is based on the BLIP2 (Li et al., 2023) implementation and pretrained weight. We will publicly release our codes and pretrained model upon acceptance.

ETHICS STATEMENT

The HS-FashionIQ dataset was annotated by 61 human annotators, and received IRB approval. We ensured participant anonymity and do not retain any personal data beyond payment information, which will be deleted later.

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

# A ALGORITHM

---

**Algorithm 1** Training Flow of *CIRQRS*

---

Inputs: Parameters $\theta$, Dataset $\mathbb{D}$, Candidate Images $\mathbb{I}$, Number of defining negative set $n_{def}$, Initial negative set size $n_{neg}$, Total epochs $n_{epoch}$

1: **for** each epoch $e$ **do**
2:  **if** $e == 0$ **then** $\quad\quad\quad\quad\quad\quad\quad\quad$ ▷ Warmup by defining negative set as whole candidates
3:   $\mathcal{S}_{\text{neg}} \leftarrow \mathbb{I} \setminus y_I$
4:  **else if** $e > 0$ and $e \mod \lfloor n_{epoch}/n_{\text{neg}} \rfloor == 0$ **then** $\quad$ ▷ Define negative set using *CIRQRS*
5:   $\mathcal{S}_{\text{neg}} \leftarrow \texttt{DefineNegativeSet}(\theta, \mathbb{I}, \mathbb{D}, n_{\text{neg}})$ $\quad\quad\quad\quad$ ▷ Section (3.3)
6:   $n_{neg} \leftarrow n_{neg}//2$
7:  **end if**
8:  **for** each batch $b$ **do**
9:   $I_{n_b} \leftarrow \texttt{RandomSample}(\mathcal{S}_{\text{neg}})$ $\quad\quad\quad\quad$ ▷ Randomly sample one negative per query
10:   $\mathcal{L}'_{match} \leftarrow -\log(\sigma(s(x_{I_b}, x_{T_b}, I_{p_b}) - s(x_{I_b}, x_{T_b}, I_{n_b})))$ $\quad\quad$ ▷ Equation (5)
11:   $\theta \leftarrow \theta - \eta\nabla_\theta\mathcal{L}'_{match}$
12:  **end for**
13: **end for**

---

# B EVIDENCES OF DEFINING HARD NEGATIVES

## B.1 QUANTITATIVE EVIDENCE

We conducted experiment with FashionIQ dataset, which compared two different versions of training. (1) *CIRQRS*, which defines the negative set as the top $n_{neg}$ images that have *lower score than the target*, (2) INCLUDE, which defines the negative set as the top $n_{neg}$ images. Here, INCLUDE indicates the case where relevant (false-negative) image is selected as a negative. Figure 6 shows that INCLUDE drastically ruined the model training, indicating that images with higher *CIRQRS* are likely to be highly relevant.

Note that this experiment is similar to the setting with Figure 3. However, previously we set $n_{neg}$ to 1000 which yields the best performance. In this experiment, to directly observe the impact of selecting image with higher *CIRQRS* than the target as a negative, we set $n_{neg}$ to 50, matching the configuration of the original *CIRQRS* training.

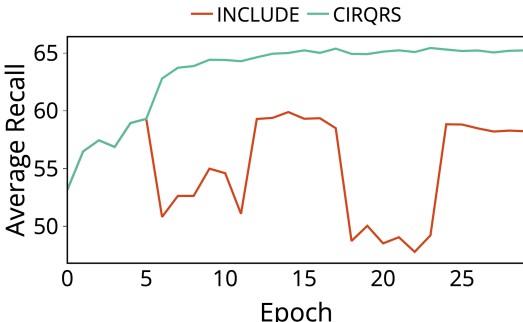

**Figure 6:** Result of three different training strategies based on how to define the negative set : (1) *CIRQRS*, as the highest score of $n_{neg}$ images with lower than the target, (2) INCLUDE, as the highest score of $n_{neg}$ images.

## B.2 QUALITATIVE EVIDENCE

We conducted qualitative experiments using the FashionIQ dataset, specifically the 'shirt' and 'dress' categories, to visually analyze our approach. Using trained *CIRQRS*, we extracted images

with higher score than the target as well as those with the lowest scores. As shown in Figures 7 and 8, images with scores higher than the target are often relevant to the query, indicating they should not be included in the negative set. Additionally, images with the lowest scores are highly irrelevant to the query, which would result in suboptimal training if included in the negative set. Note that in the 4th row of Figure 7 and the 2nd and 4th rows of Figure 8, fewer than five images are shown because only 4, 2, and 4 images, respectively, scored higher than the target.

# C  HS-FASHIONIQ DATASET

## C.1  DATA STATISTICS OF HS-FASHIONIQ

Figure 9 shows the relevance score statistics from human annotations in the HS-FashionIQ dataset. A total of 3,050 queries and 6,100 sets of retrieved images were annotated with corresponding relevance scores.

## C.2  DATA ANNOTATION EXAMPLES

We conducted a user survey via Google Forms. Each form consisted of instructions and 25 questions, with each question including a query and two different sets of retrieved images. Each participant completed two forms, covering 50 queries from the FashionIQ validation dataset, with no overlap between participants. Fig 10 shows the guidelines provided to participants, who were asked to score the retrieved image sets based on relevance to the query using a 5-point Likert scale. Figure 12 illustrates an example of a reference image, relative text, and two sets of retrieved images from different CIR models. Finally, (1) participants rated the relevance of each set, (2) indicated which set they preferred, and (3) noted whether any results were irrelevant to the reference image or text, as shown in Figure 11.

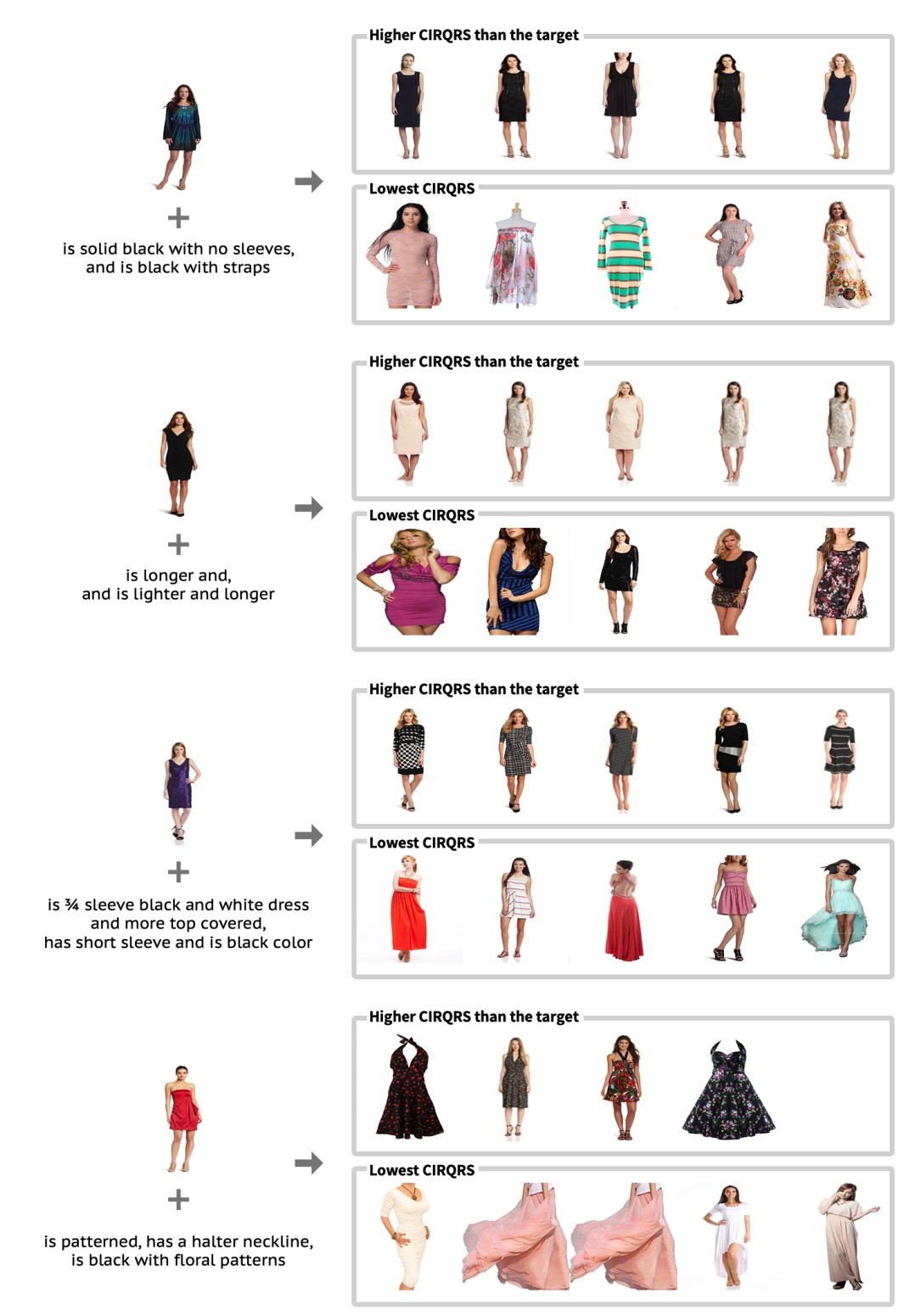

**Figure 7:** Qualitative analysis with FashionIQ dress showing that images with higher *CIRQRS* scores are likely false negatives, which should be excluded from the negative set, while images with the lowest *CIRQRS* scores are overly irrelevant and noisy.

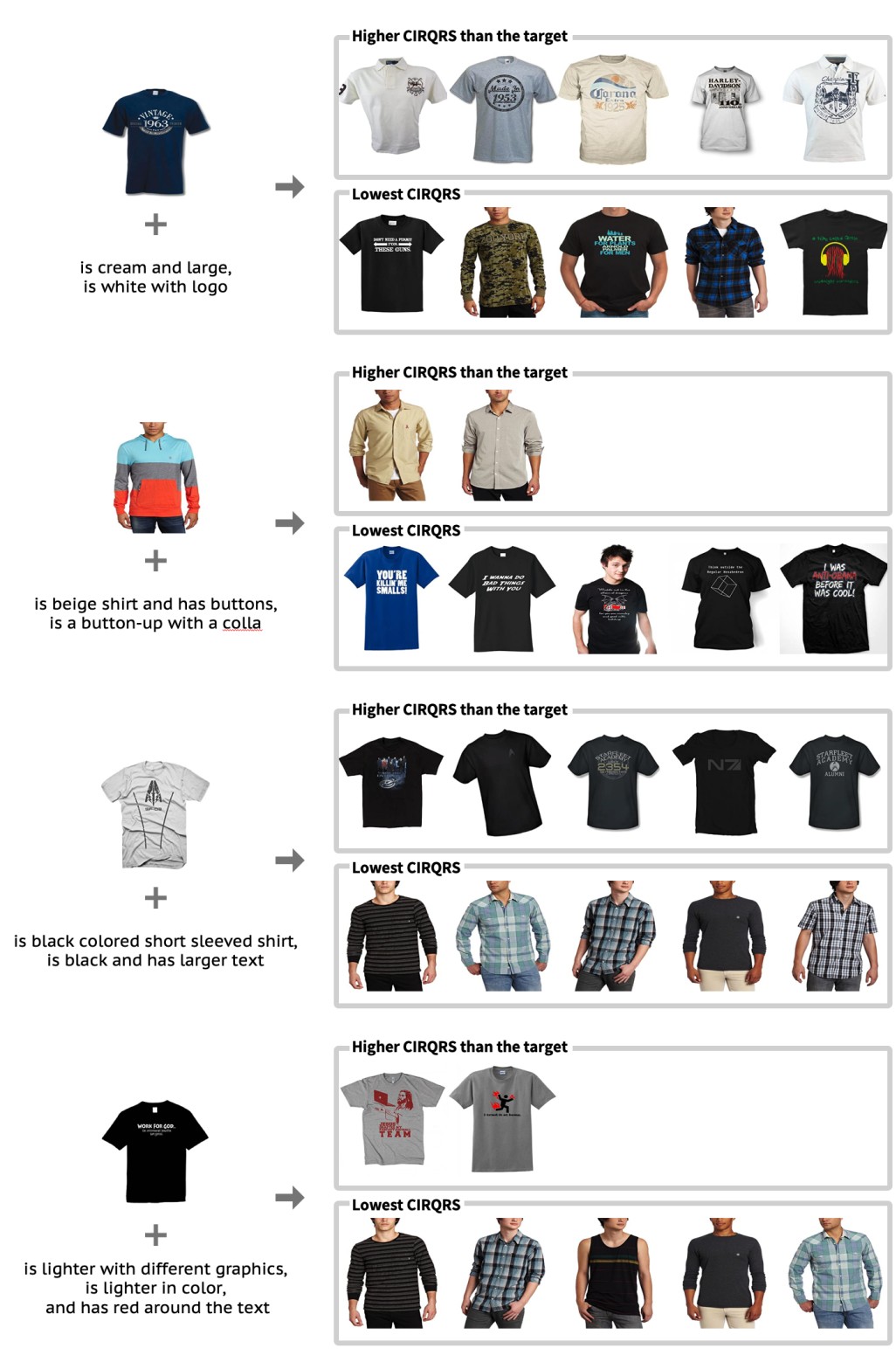

**Figure 8:** Qualitative analysis with FashionIQ shirt showing that images with higher *CIRQRS* scores are likely false negatives, which should be excluded from the negative set, while images with the lowest *CIRQRS* scores are overly irrelevant and noisy.

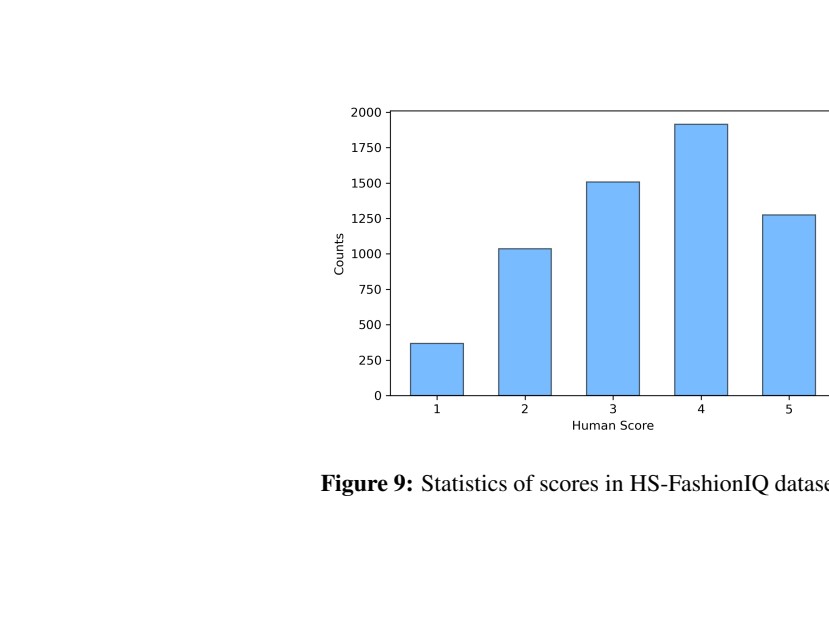

**Figure 9:** Statistics of scores in HS-FashionIQ dataset.

1. Original Image: The image you have.
2. User Text: A description of how the image you are looking for differs from the original image.
3. Two sets of image search results: Each set contains 5 images searched through two different shopping malls.

✅ Evaluation Method:
Please evaluate how well the two sets of image search results match the original image and the user text. Each set consists of 5 images, and the higher the match with the original image and the user text, the higher the score you should give. You should evaluate each set as a whole, not each individual image.

1: Does not match at all.
2: Does not match.
3: Matches to an average degree.
4: Matches.
5: Matches very well.

✅ Notes:
- Recommend completing the survey on a PC rather than a mobile device to make it easier to view the images.

- The order of the 5 images within the image search result sets does not matter.

- The search results are not generated images but are found from a fixed set of images that best match the original image and user text. Therefore, the quality of the search results may not meet user expectations. Please score as consistently as possible.

- The user text in this survey is from the dataset as is, so there may be typos or duplicated expressions.

- There are questions throughout the survey where you will be asked to explain the reason for your score.

- There will be simple math problems before proceeding to the next question. If you get these problems wrong, you cannot move on to the next page.

**Figure 10:** Guidelines for user survey.

Score for shopping mall 1 *

1: Does not match at all, 2: Does not match, 3: Matches to an average degree, 4: Matches, 5: Matches very well

| | 1 | 2 | 3 | 4 | 5 |
|---|---|---|---|---|---|
| | ○ | ○ | ○ | ○ | ○ |

Score for shopping mall 2 *

1: Does not match at all, 2: Does not match, 3: Matches to an average degree, 4: Matches, 5: Matches very well

| | 1 | 2 | 3 | 4 | 5 |
|---|---|---|---|---|---|
| | ○ | ○ | ○ | ○ | ○ |

Preference Question *

Which shopping mall's results do you prefer?

☐ Shopping Mall 1

☐ Shopping Mall 2

(Optional) Irrelevance Check Question

☐ Two set of search results are irrelevant to original image or user text

**Figure 11:** Example of questions in user survey.

Original Image / User Text: is more green and has 3/4 sleeves and has more colors and is sportier

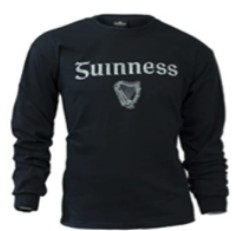

Shopping Mall 1

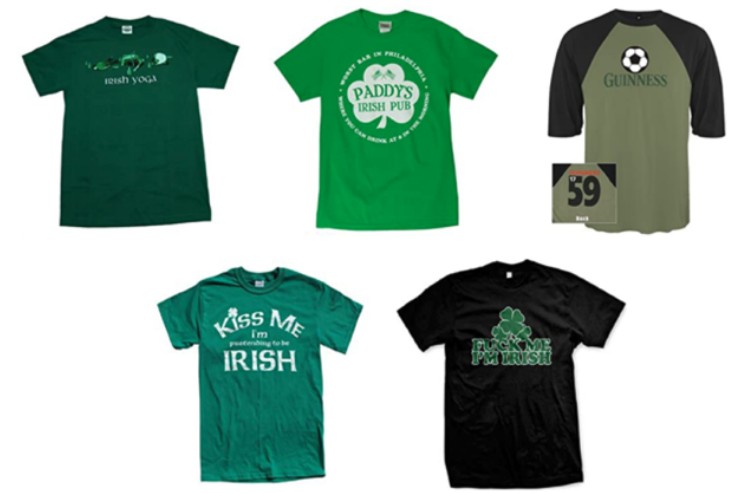

Shopping Mall 2

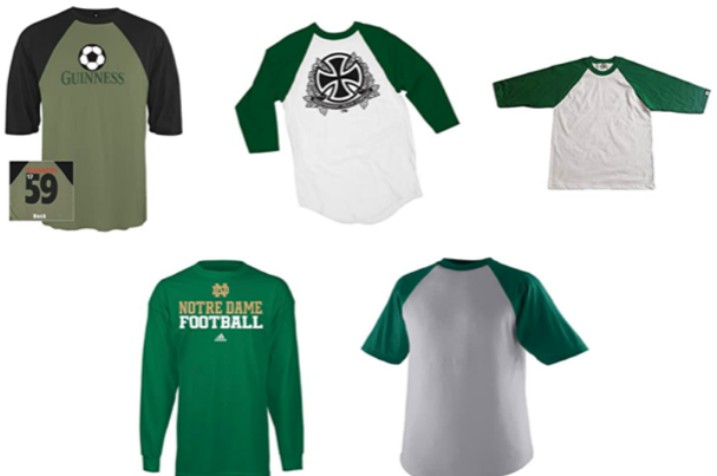

**Figure 12:** Example of query and two set of retrieved images in user survey.

