# OpenReview forum: "CIRQRS: Evaluating Query Relevance Score in Composed Image Retrieval"
_ICLR.cc/2025/Conference — Submitted to ICLR 2025_

### Official Review · Reviewer_1H4V · 2024-10-30

**Soundness:** 2
**Presentation:** 2
**Contribution:** 2
**Rating:** 5
**Confidence:** 4

**Summary:**

The paper introduces a new evaluation metric for Composed Image Retrieval (CIR) called CIRQRS (Composed Image Retrieval Query Relevance Score), aimed at overcoming limitations of the commonly used Recall@k metric. The authors offer a new dataset HS-FashionIQ, which scores different candidate groups of images of a few CIR models. The authors shows SoTA performance in most metrics of the CIRR and FashionIQ datasets.

**Strengths:**

- The idea of relevance score in image retrieval is good, especially human preference of retrieved results.
- Authors show SoTA performance on the main CIR datasets
- Training procedure involving hard-negatives is shown in the paper, suggest a better training paradigm to CIR.

**Weaknesses:**

General:

- Line 037: The comment “CIR provides a more intuitive way to express search intent” - seems subjective. While CIR does enable a multimodal query format (image + text), which can simplify certain cases (e.g., describing a specific T-shirt design that may be hard to convey in words alone), using text as a query is often more intuitive for me (text-to-image retrieval task). I recommend rephrasing this sentence to clarify the intended advantage.
- Line 042: The example provided is helpful, but it would be especially beneficial to include the ground truth target image for clarity. This is important for readers who may not be familiar with the dataset.
- Related Work:  It would be more logical to introduce the “Vision-Language Foundation Model” paragraph first, as it provides the foundation for most CIR-related studies.
- Line 227: The claim that “as images with higher CIRQRS than the target are likely to be highly relevant after sufficient training” needs support. Consider using “Precision@K” labeling, accounting for false negatives (images that match but are not labeled as such). Otherwise, higher scores than the GT image could just be noise or indicate an imperfect model.
- Line 262 refers to Figure 5: This example is unclear. What distinguishes set A from set B? Why do the same images have different CIRQRS scores across sets? What is the GT image, and what do the User Preference symbols (v or x) mean? This example should clarify the annotation process (line 262), but it currently feels confusing.
- The scores CIRQRS-model outputs, to my understanding, are just similarity scores as any other casual CIR model produce. The authors shows SoTA performance, in some metric, with their different training procedure that considers “hard-negative” samples. The authors claim to introduce a new evaluation metric that “overcomes the limitation of Recall@k” (line 479), but in practice, all different model in the paper was evaluated using the Recall@K metric and not the new one.

Annotations:
- I have a major concern about the annotation process. Image Retrieval (IR) datasets often include many “false-negative” examples—images labeled as “negative” by default but which humans would consider “positive.” Ideally, a human annotator would review the entire image corpus for each query to mark each image as positive or negative, but that’s impractical. While CIRQRS score annotations are valuable for the community, I worry they may still include “false-negatives,” where more positives are marked per sample. I suggest an extra step to label images in the candidate set as positive or negative, alongside CIRQRS score, to better distinguish true positives from “relevant but negative” images.

Evaluation:
 - Tables 4 and 5 compare different methods to the paper results (CIRQRS). As described in Sections 3 and 5, the authors leveraged the BLIP-2 foundation model for the the CIR task. Despite being trained differently, it is essential to present the BLIP-2 baseline performance on these datasets, compared to current CIRQRS. It will clarify from where the improvement in results comes from: theoretically, one may suspect that the BLIP-2 model (that shown to be a strong backbone) may outperform the CIRQRS baseline which maybe only decreases the result of its backbone. On the other hand, the CIRQRS baseline may improve BLIP-2 baseline in a large margin, due to a better training procedure. It is an essential comparison to make here, which is missing in the evaluation tables.

- Just to make it clear: does the authors method rely solely on the FashionIQ training set, or it also trained on HQ-FashionIQ?
* how beneficial would it be to train on both HQ-FashionIQ and FashionIQ? Since your method selects “hard negatives” using the scoring model (Sec 3.2), could adding HQ-FashionIQ data improve the Recall@K metrics?

Evaluation on HS-FASHIONIQ:

- In the paper, “CIRQRS” is referred as a score, metric and as a model/method (based on BLIP-2)..  Please clarify this usage in context or consider renaming (e.g., “CIRQRS-Model”).
Specifically, in Sec 5.2, you compare Recall@K and CIRQRS metrics to human-preferences. What model was used for these results? What is the actual values of these numbers on this model, and how it changes across different models?
It is not fully clear to me.

Correlation with Human Score (Sec 5.2):

I’m not convinced this experiment is necessary. Different metrics assess different things: Recall@K measures how many GT targets appear in the top K, while human preference between two sets of results measures, indeed, human preference, and takes into account only top-k results. Imagine a bad CIR model that leaves all GT targets least in the results, but still top ranks “relevant” images based on shallow chartersitic such color - in this case, the Recall@K should be very low, while human preference will be high. It would be more “comparable” to conduct this experiment with Precision@K metric (but, it is not widely used as Recall@K).


I think the paper idea is good and important, but the flow is not entirely clear. The authors shall distinguish between method, metric and score, and provide more evidence for some claims. For example, since the CIRQRS metric presented as the main contribution in the paper, it is essential to include a substantial comparison of CIRQRS values across several CIR models. Without this, it is unlikely other CIR studies would adopt this metric, which would limit the paper’s impact.

**Questions:**

1. Eq. 1: “the inner product of the query and image embeddings” - I assume you mean “the inner product of the image embeddings of the query and the candidate image.” Is that correct?

---

> ### Author Response · Authors · 2024-11-18
> **Response to Reviewer 1H4V - W1, W2, W3**
>
> We deeply appreciate your insightful comments and efforts in reviewing our manuscript. We marked our revisions in the manuscript with “blue” fonts. Here, we respond to each of your comments.
>
> **W1**
>
> > Line 037: The comment “CIR provides a more intuitive way to express search intent” - seems subjective. While CIR does enable a multimodal query format (image + text), which can simplify certain cases (e.g., describing a specific T-shirt design that may be hard to convey in words alone), using text as a query is often more intuitive for me (text-to-image retrieval task). I recommend rephrasing this sentence to clarify the intended advantage.
>
> **A1**
>
> Thank you for your detailed review. We agree that text could be more intuitive than the multimodal query in some cases. We recognize that our original wording could lead to misinterpretation, so we have revised it in our updated manuscript to: “CIR enhances search precision, particularly in cases where describing visual details is challenging with text alone, making it valuable for applications in e-commerce and internet search.”
>
>
> **W2**
>
> > Line 042: The example provided is helpful, but it would be especially beneficial to include the ground truth target image for clarity. This is important for readers who may not be familiar with the dataset.
>
> **A2**
>
> Thank you for your suggestion. We have added the ground truth target image for the example in the first row in our revised manuscript.
>
>
> **W3**
>
> >Related Work: It would be more logical to introduce the “Vision-Language Foundation Model” paragraph first, as it provides the foundation for most CIR-related studies.
>
> **A3**
>
> Thank you for your suggestion. In our revised manuscript, we have changed the order of the related works.

---

> ### Author Response · Authors · 2024-11-18
> **Response to Reviewer 1H4V - W4, W5**
>
> **W4**
>
> > Line 227: The claim that “as images with higher CIRQRS than the target are likely to be highly relevant after sufficient training” needs support. Consider using “Precision@K” labeling, accounting for false negatives (images that match but are not labeled as such). Otherwise, higher scores than the GT image could just be noise or indicate an imperfect model.
>
> **A4**
>
> Thank you for pointing this out. We agree that evidence is essential to support our method’s core approach, where we exclude images with higher CIRQRS than the target when defining the negative set. Since CIR datasets only have a few (mostly one) ground truths per query and false-negative images are not annotated, calculating Precision@K is challenging. However, we do agree that multiple images may be relevant to the query, which is the primary issue that our paper aims to solve. CIRQRS considers each image's relevancy to evaluate each set, while Recall@K simply evaluates whether the target image is present or not in the retrieved set.
>
> Line 227 shows our intuition on selecting ‘hard images’, which stems from our training objective, where we expect relevant images to achieve scores similar to or higher than the target. While theoretically demonstrating this effect is challenging due to the stochastic nature of deep learning and the lack of annotated relevant images, we conducted additional experiments comparing two different versions of the training. (1) CIRQRS, which defines the negative set as the top $n_{neg}$ images with a lower score than the target, (2) INCLUDE, which defines the negative set as the top $n_{neg}$ images. (Including the images that have a higher score than the target).
>
> We have added the results graph, Figure 6, in Appendix B.1, showing that INCLUDE drastically ruined the model training, indicating that images with higher CIRQRS are likely to be highly relevant.
>
> To further support this statement, we added the qualitative examples in Appendix B.2, demonstrating that images with higher CIRQRS than the target are highly likely to be relevant.
>
>
> **W5**
>
> > Line 262 refers to Figure 5: This example is unclear. What distinguishes set A from set B? Why do the same images have different CIRQRS scores across sets? What is the GT image, and what do the User Preference symbols (v or x) mean? This example should clarify the annotation process (line 262), but it currently feels confusing.
>
> **A5**
>
> Thank you for pointing this out. We agree that it needs more explanation in Section 4, as we refer to Figure 5. In the revised manuscript, we have added the explanation in the caption of Figure 5, and below is a further explanation.
>
> Figure 5  is one example data point from our HS-FashionIQ dataset. Following the explanation from Section 4, each data point consisted of a query with 2 sets of retrieved images from 2 random CIR models. Each annotator then was asked to __rate each set__ of images and __their preferred set__.
>
> Set A and Set B are the retrieved images related to the query, and since both are retrieved from different models, both contain different sets of images. If you look closely, indeed, the same images have the same CIRQRS score, even though the order of the images is different. The GT image is highlighted by a red box. The user preference symbol denotes the set the annotator prefers, as described above and in Section 4.

---

> ### Author Response · Authors · 2024-11-18
> **Response to Reviewer 1H4V - W6**
>
> **W6-1**
>
> > The scores CIRQRS-model outputs, to my understanding, are just similarity scores as any other casual CIR model produce. The authors shows SoTA performance, in some metric, with their different training procedure that considers “hard-negative” samples.
>
> **A6-1**
>
> Indeed, similarity scores from other CIR models could also serve as scores. However, since these models are not specifically designed to assign higher scores to relevant images, their primary goal remains to rank a single target image as high as possible. Following your suggestion, we conducted an experiment where we treated similarity scores from other CIR models as scores and compared them with our CIRQRS against the HS-FashionIQ dataset to evaluate their alignment with human preferences. The table shows that our CIRQRS has the strongest alignment compared to other baselines.
>
> | Methods | Preference Rate |
> |------------|-----------------|
> | CLIP4CIR | 0.6608 |
> | BiBLIP4CIR | 0.6700 |
> | CoVRBLIP | 0.7276 |
> | SPRC | 0.7339 |
> | __CIRQRS__ | __0.7524__ |
>
> **W6-2**
>
> > The authors claim to introduce a new evaluation metric that “overcomes the limitation of Recall@k” (line 479), but in practice, all different model in the paper was evaluated using the Recall@K metric and not the new one.
>
> **A6-2**
>
> Thank you for your suggestion. We evaluated the four baseline models (CLIP4CIR, BLIP4CIR, CoVR-BLIP, and SPRC) on two representative datasets, FashionIQ and CIRR, using our metric CIRQRS.
>
> | FashionIQ | Dress CIRQRS@10 | Dress CIRQRS@50 | Shirt CIRQRS@10 | Shirt CIRQRS@50 | TopTee CIRQRS@10 | TopTee CIRQRS@50 | Avg. CIRQRS@10 | Avg. CIRQRS@50 | Average |
> |:----------:|-----------------|-----------------|-----------------|-----------------|------------------|------------------|----------------|----------------|---------|
> | CLIP4CIR | 67.97 | 66.35 | 68.61 | 66.51 | 67.62 | 65.88 | 68.07 | 66.25 | 67.16 |
> | BiBLIP4CIR | 71.03 | 68.86 | 69.90 | 67.51 | 70.16 | 67.64 | 70.36 | 68.00 | 69.18 |
> | CoVRBLIP | 72.07 | 69.78 | 71.41 | 68.76 | 71.38 | 68.64 | 71.62 | 69.06 | 70.34 |
> | SPRC | 72.26 | 69.99 | 71.89 | 69.27 | 71.76 | 69.07 | 71.97 | 69.44 | 70.71 |
>
> | CIRR | CIRQRS@1 | CIRQRS@5 | CIRQRS@10 | CIRQRS@50 | CIRQRS s@1 | CIRQRS s@2 | CIRQRS s@3 | Mean 5 + 1 | Mean |
> |:----------:|----------|----------|-----------|-----------|------------|------------|------------|------------|-------|
> | CLIP4CIR | 69.42 | 67.92 | 66.89 | 63.38 | 74.78 | 72.86 | 71.57 | 71.35 | 69.55 |
> | BiBLIP4CIR | 68.70 | 66.74 | 65.50 | 61.62 | 73.11 | 71.96 | 71.07 | 69.92 | 68.39 |
> | CoVRBLIP | 76.08 | 72.80 | 71.07 | 66.04 | 77.20 | 74.27 | 72.32 | 75.00 | 72.83 |
> | SPRC | 75.29 | 72.31 | 70.69 | 65.85 | 76.89 | 74.12 | 72.26 | 74.60 | 72.49 |
>
> We calculated the CIRQRS@k using mean CIRQRS of the retrieved set (Equation 7)
>
>
> Both tables show that CIRQRS@K decreases as K increases, indicating that all CIR methods retrieve more irrelevant images as K grows. In the FashionIQ dataset, SPRC achieved the best performance, following the same trend observed in Table 4 of our manuscript. However, in the CIRR dataset, the trends differ from those in Table 5, CLIP4CIR outperforms Bi-BLIP4CIR, and CoVRBLIP surpasses SPRC. This suggests that, although Bi-BLIP4CIR and SPRC achieved higher Recall@K, they are less aligned with human preferences than CLIP4CIR and CoVRBLIP.
>
> As you can see in the A6-1, the four baselines follow the same order as CIRQRS in the FashionIQ dataset, perfectly reflecting the trend observed in the CIRQRS table for FashionIQ. However, we haven’t created the Human-scored CIRR dataset, making it challenging to explain why CLIP4CIR and CoVRBLIP perform better on the CIRR dataset with the CIRQRS. We plan to create a human-scored dataset for not only the FashionIQ dataset but also CIRR, aiming to further advance the CIR community.

---

> ### Author Response · Authors · 2024-11-18
> **Response to Reviewer 1H4V - W7, W8**
>
> **W7**
>
> > I have a major concern about the annotation process. Image Retrieval (IR) datasets often include many “false-negative” examples—images labeled as “negative” by default but which humans would consider “positive.” Ideally, a human annotator would review the entire image corpus for each query to mark each image as positive or negative, but that’s impractical. While CIRQRS score annotations are valuable for the community, I worry they may still include “false-negatives,” where more positives are marked per sample. I suggest an extra step to label images in the candidate set as positive or negative, alongside CIRQRS score, to better distinguish true positives from “relevant but negative” images.
>
> **A7**
>
> We agree that there might be multiple images that are relevant to the query, and this is the main problem that we aim to solve in our paper, in which we create a metric that considers each image's relevance score. However, in our HS-FashionIQ dataset, we employ human annotators to evaluate the relevance and preference of each retrieved image set. In particular, the human annotator __does not__ annotate whether a single image is positive or negative. Our main intention for this dataset is to evaluate our new metric CIRQRS and future CIR models in terms of human preferences.
>
> **W8**
>
> > Tables 4 and 5 compare different methods to the paper results (CIRQRS). As described in Sections 3 and 5, the authors leveraged the BLIP-2 foundation model for the the CIR task. Despite being trained differently, it is essential to present the BLIP-2 baseline performance on these datasets, compared to current CIRQRS. It will clarify from where the improvement in results comes from: theoretically, one may suspect that the BLIP-2 model (that shown to be a strong backbone) may outperform the CIRQRS baseline which maybe only decreases the result of its backbone. On the other hand, the CIRQRS baseline may improve BLIP-2 baseline in a large margin, due to a better training procedure. It is an essential comparison to make here, which is missing in the evaluation tables.
>
> **A8**
>
> We have presented the performance of BLIP-2 in the table below on FashionIQ and CIRR datasets. As we can see, there is a massive performance increase when we train using our method. We can also see this trend in Figure 3, where performance improves dramatically during training.
>
> |     FashionIQ    | Dress CIRQRS@10 | Dress CIRQRS@50 | Shirt CIRQRS@10 | Shirt CIRQRS@50 | TopTee CIRQRS@10 | TopTee CIRQRS@50 | Avg. CIRQRS@10 | Avg. CIRQRS@50 | Average   |
> |:----------:|-----------------|-----------------|-----------------|-----------------|------------------|------------------|----------------|----------------|-----------|
> | BLIP-2     |            7.09 |           17.30 |            5.59 |           11.43 |             9.38 |            17.39 |           7.36 |          15.37 |     11.37 |
> | **CIRQRS** |       **49.33** |       **73.33** |       **56.97** |       **74.58** |        **59.00** |        **78.23** |      **55.10** |      **75.38** | **65.24** |
>
> |    CIRR    | Recall@1  | Recall@5  | Recall@10 | Recall@50 | Recall s@1 | Recall s@2 | Recall s@3 | Mean 5 + 1 | Mean      |
> |:----------:|-----------|-----------|-----------|-----------|------------|------------|------------|------------|-----------|
> | BLIP-2     | 11.831    | 31.181    | 43.157    | 66.313    | 42.602     | 65.301     | 80.337     |      36.89 |     48.67 |
> | **CIRQRS** | **53.42** | **83.81** | **91.21** | **98.53** |  **79.78** |  **92.36** |  **96.87** |  **81.80** | **85.14** |
>
> We further compared it on the HS-FashionIQ dataset, and the results show that CIRQRS also increased the alignment with human preferences compared to the BLIP-2 model.
>
> | Methods | Preference Rate |
> |---------|-----------------|
> | BLIP-2 | 0.6175 |
> | **CIRQRS** | **0.7524** |

---

> > ### Comment · Reviewer_1H4V · 2024-11-21
> >
> > I appreciate the authors’ thorough response. Regarding A8, my understanding is that you compared BLIP-2 in a zero-shot setting with fine-tuned CIRQRS (on CIRR/FashionIQ). However, my main intent was to explore the impact of the backbone during fine-tuning. To clarify, my primary question is: “Is the CIRQRS method better than simply fine-tuning BLIP-2?” I hope this provides additional clarity and underscores the significance of conducting this experiment to demonstrate why CIRQRS is a superior approach compared to "just" fine-tuning a more robust backbone.
> >
> > For A12, I apologize, but I still find the explanation unclear. Different metrics capture different aspects. While I understand the problem addressed in your paper, I am not convinced of the relevance of comparing Recall@K—which evaluates “how many ground truths are left behind”—with human preference, which reflects “how satisfied users are with the presented results.” In my view, the argument that the CIRQRS metric correlates more closely with human preference than Recall@K does not sufficiently validate its value. Instead, demonstrating that CIR models achieving higher human preference scores also achieve higher CIRQRS scores would more effectively establish the importance of the CIRQRS metric.

---

> > > ### Author Response · Authors · 2024-11-22
> > > **Response to Reviewer 1H4V's update**
> > >
> > > Thank you for taking the time to re-evaluate our work and for raising the score. We greatly appreciate your constructive feedback, which has helped us clarify and improve our submission.
> > >
> > > > Is the CIRQRS method better than simply fine-tuning BLIP-2?
> > >
> > > We sincerely appreciate your follow-up questions and are glad to provide further clarification. CIRQRS fine-tuned the pre-trained BLIP-2 model using our training objectives, including the scoring model and the negative set definition described in Section 3. We regret not explicitly stating that we used a "pre-trained" BLIP-2 model and apologize for this omission. If you understood as CIRQRS was trained from scratch rather than fine-tuned from BLIP-2, we will revise the manuscript to provide a clearer explanation.
> > >
> > > Moreover, the previous state-of-the-art CIR method, SPRC, is also one of the methods that fine-tuned BLIP-2 using its own training objectives. However, as shown in A6-1, CIRQRS achieves better alignment with human preferences on the HS-FashionIQ dataset and achieves state-of-the-art performance when applied as a CIR model on FashionIQ and CIRR datasets (denoted as CIRQRS-Model in Table 4 and 5), highlighting CIRQRS as the current superior approach.
> > >
> > > > Do CIR models with higher human preference scores also achieve higher CIRQRS scores?
> > >
> > > Thank you for your concerns and constructive suggestions. We agree that it would be ideal to directly verify that “CIR model achieving higher human preference scores also achieves higher CIRQRS scores”. However, this is challenging due to limitations in existing datasets with the following detailed reasons.
> > >
> > > First, extracting human preference scores requires datasets like HS-FashionIQ, where human annotators label their preferred set of retrieved images. Unfortunately, widely used CIR evaluation datasets such as FashionIQ and CIRR lack such annotations.
> > >
> > > Second, calculating a specific CIR model’s CIRQRS score requires the model to retrieve images for a given query. However, the current HS-FashionIQ dataset consists of two sets of retrieved images per query, which are randomly selected from the four different CIR models. This means HS-FashionIQ does not provide retrieval results exclusively from a single model, making direct comparisons challenging.
> > >
> > > To indirectly validate the effectiveness of CIRQRS, we conducted the following steps:
> > >
> > > 1. Using the FashionIQ dataset:
> > > - (a) CIR models retrieve images, and calculate the CIRQRS scores of these retrieved images.
> > > - (b) The CIR models were ranked based on their CIRQRS scores.
> > > 2. Using the HS-FashionIQ dataset:
> > > - (a) The same CIR models were evaluated using human preference scores.
> > > 3. The rankings from CIRQRS scores (step 1b) were compared to the rankings from human preference scores (step 2a) to verify alignment.
> > >
> > > For example, evaluating four CIR models (CLIP4CIR, Bi-BLIP4CIR, CoVR-BLIP, and SPRC) using our metric CIRQRS on the FashionIQ dataset yields the following results:
> > >
> > > | FashionIQ | Dress CIRQRS@10 | Dress CIRQRS@50 | Shirt CIRQRS@10 | Shirt CIRQRS@50 | TopTee CIRQRS@10 | TopTee CIRQRS@50 | Avg. CIRQRS@10 | Avg. CIRQRS@50 | Average |
> > > |:--------:|-----------------|-----------------|-----------------|-----------------|------------------|------------------|----------------|----------------|---------|
> > > | CLIP4CIR | 67.97 | 66.35 | 68.61 | 66.51 | 67.62 | 65.88 | 68.07 | 66.25 | 67.16 |
> > > | BiBLIP4CIR | 71.03 | 68.86 | 69.90 | 67.51 | 70.16 | 67.64 | 70.36 | 68.00 | 69.18 |
> > > | CoVRBLIP | 72.07 | 69.78 | 71.41 | 68.76 | 71.38 | 68.64 | 71.62 | 69.06 | 70.34 |
> > > | SPRC | 72.26 | 69.99 | 71.89 | 69.27 | 71.76 | 69.07 | 71.97 | 69.44 | 70.71 |
> > >
> > > The results indicate that the ranking of CIR models based on CIRQRS is as follows: **SPRC > CoVR-BLIP > Bi-BLIP4CIR > CLIP4CIR**
> > >
> > > Next, we used the HS-FashionIQ dataset to extract the human preference rate of CIR models, as shown below:
> > >
> > > | Methods | Preference Rate |
> > > |------------|-----------------|
> > > | CLIP4CIR | 0.6608 |
> > > | BiBLIP4CIR | 0.6700 |
> > > | CoVRBLIP | 0.7276 |
> > > | SPRC | 0.7339 |
> > >
> > > The results show that the ranking order of CIR models with the CIRQRS scores on the FashionIQ dataset aligns with the human preference rates observed on the HS-FashionIQ dataset **(SPRC > CoVR-BLIP > Bi-BLIP4CIR > CLIP4CIR)**. Since HS-FashionIQ was created from a subset of the FashionIQ validation set, the data distributions are comparable. We believe these findings suggest that “CIR model that achieves a higher CIRQRS score is likely to achieve higher human preference rate”, supporting the validity of CIRQRS as a reliable evaluation metric.
> > >
> > > We hope our response addresses your question. Feel free to let us know if you have any further questions.

---

> ### Author Response · Authors · 2024-11-18
> **Response to Reviewer 1H4V - W9, W10, W11, W12, Q1**
>
> **W9**
>
> > Just to make it clear: does the authors method rely solely on the FashionIQ training set, or it also trained on HQ-FashionIQ?
>
> **A9**
>
> We __did not__ use HS-FashionIQ for training and only used the training split for FashionIQ and CIRR to train our models. We strictly use HS-FashionIQ as evaluation data as it’s created from the validation set of FashionIQ.
>
> **W10**
>
> > how beneficial would it be to train on both HQ-FashionIQ and FashionIQ? Since your method selects “hard negatives” using the scoring model (Sec 3.2), could adding HQ-FashionIQ data improve the Recall@K metrics?
>
> **A10**
>
> HS-FashionIQ is data created from the validation split of FashionIQ, and training using this data will introduce data leakage between the training and validation set. However, in broader terms, HS-FashionIQ is a set of images retrieved from other CIR models, and it might be used to treat it as the ‘hard negative’ set. But practically, utilizing a dataset like this for training is not efficient as we have to do inferences from multiple CIR models.
>
> **W11**
>
> > In the paper, “CIRQRS” is referred as a score, metric and as a model/method (based on BLIP-2).. Please clarify this usage in context or consider renaming (e.g., “CIRQRS-Model”). Specifically, in Sec 5.2, you compare Recall@K and CIRQRS metrics to human-preferences. What model was used for these results? What is the actual values of these numbers on this model, and how it changes across different models? It is not fully clear to me.
>
> **A11**
>
> Thank you for the suggestion. In our current paper, we used CIRQRS interchangeably for both the metric and the model, which we agree could cause confusion. We have updated our manuscript in Section 5.2 to use the term "CIRQRS-Model" when referring to its role in retrieving images and comparing it with existing CIR models. This change provides a clearer distinction between its function as a metric and as a model.
>
> Each query in the HS-FashionIQ dataset has two distinct sets of retrieved images generated by two randomly selected CIR models from CLIP4CIR, Bi-BLIP4CIR, CoVR-BLIP, and SPRC. In Section 5.2, CIRQRS for each set is calculated using Equation 7 (average CIRQRS of the images), while Recall@5 is based on whether the target image appears in the set. Human scores are provided by annotators based on the set's relevance to the query, and annotators also label their preferred set. The question posed to annotators is detailed in Appendix C, Figure 11.
>
> Table 2 shows the Spearman correlation between the human score from human annotation and the score obtained by metrics Recall@K and CIRQRS. In Table 3, we show the preference rate, which is the conditional probability that a set is preferred, given that the metric score of that set is higher (Equation 8 in our paper). The result from Tables 2 and 3 shows that CIRQRS shows a stronger correlation and alignment than Recall@K
>
> **W12**
>
> >[Correlation with Human Score (Sec 5.2):] I’m not convinced this experiment is necessary. Different metrics assess different things: Recall@K measures how many GT targets appear in the top K, while human preference between two sets of results measures, indeed, human preference, and takes into account only top-k results. Imagine a bad CIR model that leaves all GT targets least in the results, but still top ranks “relevant” images based on shallow chartersitic such color - in this case, the Recall@K should be very low, while human preference will be high. It would be more “comparable” to conduct this experiment with Precision@K metric (but, it is not widely used as Recall@K).
>
> **A12**
>
> This example you provided is the problem we illustrated in Figure 1 and is exactly the problem we are tackling in this paper. In this work, we argue that the current widely used evaluation metric for CIR, Recall@K, is flawed due to its inability to assess the relevance of retrieved images. On top of that, current CIR datasets only have a few (mostly one) ground truths per query, making it challenging to use Precision@K. CIRQRS considers the relevance score of each image, compared to Recall@K, which only evaluates based on whether the target image is present in the retrieved set. The experiments in Section 5.2 show comparisons between CIRQRS and Recall@K when used to evaluate retrieved image sets, and the result from Tables 2 and 3 shows that CIRQRS shows a stronger correlation and alignment than Recall@K. These results indicate that human preferences are strongly influenced by the overall relevance of the images (CIRQRS) rather than the presence of the target image alone (Recall@K).
>
> **Q1**
>
> > Eq. 1: “the inner product of the query and image embeddings” - I assume you mean “the inner product of the image embeddings of the query and the candidate image.” Is that correct?
>
> **A13**
>
> No, the query embedding refers to a joint embedding of both the query text and query image.

---

> ### Comment · Reviewer_1H4V · 2024-11-25
>
> I would like to thank the authors for their effort and their detailed rebuttal.
> I raised my score from 3 to 5 following the authors' clarifications.

---

> > ### Author Response · Authors · 2024-11-26
> >
> > Thank you again for revising your score. We deeply appreciate the time and effort you dedicated to reviewing our manuscript. Your valuable feedback and constructive suggestions have significantly enhanced our work.
> >
> > We are glad that our rebuttal has resolved your concerns and will continue the discussion if you have any additional questions.
> >
> > Best regards,
> > The Authors

---

### Official Review · Reviewer_RFmP · 2024-11-02

**Soundness:** 3
**Presentation:** 3
**Contribution:** 3
**Rating:** 5
**Confidence:** 4

**Summary:**

This paper introduces CIRQRS into the CIR task, providing a user-centric evaluation metric. The HS-FashionIQ dataset with human scores was created to validate CIRQRS. The paper proposes a new strategy to refine negative images during training and improve query-based image ranking. Experiments on the CIRR and FashionIQ datasets validate the effectiveness of the proposed method.

**Strengths:**

1 ) The paper is well-written and has a clear structure. It effectively presents a novel CIR evaluation metric. The appendix includes additional details on the new dataset. \
2 ) The proposed method achieves top-ranking results, placing first and second on the FashionIQ and CIRR datasets.

**Weaknesses:**

1 ) The paper does not sufficiently highlight the novelty and effectiveness of the proposed approach. The authors should explain more to clarify the method’s advancements beyond merely introducing a new loss function. \
2 ) Since the goal is to introduce a new evaluation metric for CIR, a more detailed review of its effectiveness is needed. For example, testing CIRQRS on other models could help show its efficiency. \
3 ) In Tables 2 and 3, the paper wants to demonstrate that CIRQRS is more effective than R@5, but the process by which these data were calculated should be clarified in greater detail. Since R@5 is an important point here, it would also help to show a comparison of R@5 results on the FashionIQ dataset as in Table 4 with other methods.

**Questions:**

The paper contains minor grammatical errors. For example, on page 2, line 85, "Additionally, when CIRQRS is higher on a set of retrieved images" should use "was" instead of "is," and "limitation" on line 95 should be "limitations."

---

> ### Author Response · Authors · 2024-11-18
> **Response to Reviewer RFmP - W1**
>
> We deeply appreciate your insightful comments and efforts in reviewing our manuscript. We marked our revisions in the manuscript with “blue” fonts. Here, we respond to each of your comments.
>
> **W1**
>
> > The paper does not sufficiently highlight the novelty and effectiveness of the proposed approach. The authors should explain more to clarify the method’s advancements beyond merely introducing a new loss function.
>
> **A1**
>
> Thank you for pointing it out. We agree that we need to emphasize the novelty and effectiveness of our method more. We have added the sentences to emphasize our novelty in the method part in our revised manuscript. The key differences between CIRQRS and the existing methods are as follows.
>
> Firstly, our proposed training objective shows that representation learning can be done with preference optimization instead of the widely used[1][2] contrastive loss in CIR model training. This approach is more efficient, as each query only requires a single negative image instead of multiple negative images.
>
> However, it is not trivial to select proper negative images, as only one or few target images are annotated in the CIR dataset. Also, random selection from the corpus is suboptimal, as seen in Table 7. To address this, we propose a method to appropriately select the negative images for each training step to boost and stabilize the training with ideas stemming from self-paced learning.
>
> Previous self-paced learning works [3][4][5] (1) use __multiple__ samples as negatives, and some of them are __pre-defined__, and (2) select easy-to-hard samples as just __top K__ samples based on their criteria (e.g., loss, distance). However, in CIR, negative images are not predefined, so it is challenging to avoid false-negative images (relevant but not a target) from being selected as a negative sample. To address it, we fixed the target image as positive, which aims for images similar to the target to receive higher/similar scores as training progresses. Consequently, we define the negative set as the __top K images with scores lower than the target image__ to ensure that relevant images are excluded from selection as negatives.
>
> We conducted experiments to support this idea by comparing (1) the original CIRQRS training, (2) INCLUDE, where the top $n_{neg}$ images are defined as a negative set, like existing self-paced learning works. The result graph is shown in Appendix B.1, Figure 6. Our simple yet effective new strategy for defining the negative set enhances the training. Defining just top $n_{neg}$ images as a negative set could include the relevant image. We additionally conducted a qualitative analysis on Appendix B.2 in our revised manuscript.
>
>
>
> [1] Bai Y. et. al. Sentence-level prompts benefit composed image retrieval. In The Twelfth International Conference on Learning Representations, 2024.
>
> [2] Liu Z. et. al. Bi-directional training for composed image retrieval via text prompt learning. In Proceedings of the IEEE/CVF Winter Conference on Applications of Computer Vision,
>
> [3]Jiang, Lu, et al. "Self-paced learning with diversity." Advances in neural information processing systems 27 (2014).
>
> [4] Supancic, James S., and Deva Ramanan. "Self-paced learning for long-term tracking." Proceedings of the IEEE conference on computer vision and pattern recognition. 2013.
>
> [5] Liu, Kangning, et al. "Multiple instance learning via iterative self-paced supervised contrastive learning." Proceedings of the IEEE/CVF Conference on Computer Vision and Pattern Recognition. 2023.

---

> > ### Comment · Reviewer_RFmP · 2024-11-26
> > **Overall response**
> >
> > Thank you to the authors for their response and the detailed additional work. The explanation of CIRQRS’s innovations in negative sample selection and training stability is clear, and the supporting experiments and analysis in the table further validate its effectiveness.  However, I still suggest positioning CIRQRS more as a tool for improving training and evaluation to better highlight its practical value in CIR tasks. I have decided to maintain my score.

---

> > > ### Author Response · Authors · 2024-11-26
> > >
> > > We are glad to hear that our rebuttal has addressed your concerns. However, regarding your last suggestion, "positioning CIRQRS more as a tool for improving training and evaluation to better highlight its practical value in CIR tasks," we noticed that this point was not raised in your initial review. Could you kindly elaborate on this suggestion to help us better understand your perspective?

---

> > > > ### Author Response · Authors · 2024-12-02
> > > >
> > > > Dear Reviewer,
> > > >
> > > > We sincerely appreciate your comments and constructive feedback on our work.
> > > >
> > > > We assume that the last point you raised relates to Reviewer kfFA's suggestion regarding why CIRQRS should not be treated solely as a methodological advancement rather than as an evaluation metric. As noted in your response, CIRQRS proves its effectiveness as an evaluation metric by (1) achieving the strongest alignment with human preferences, (2) demonstrating that CIR models with higher CIRQRS scores achieve the highest human preference rates, and (3) exhibiting superior zero-shot performance, highlighting its generalizability. We believe that CIRQRS shows a critical first step toward advancing the CIR field.
> > > >
> > > > Thank you for giving us the chance to address your concerns. If these points sufficiently address your questions, we kindly request you to consider revising your rating. Please feel free to reach out if you have any additional questions or require further clarification.
> > > >
> > > > Best regards,
> > > > Authors.

---

> > > > > ### Comment · Reviewer_RFmP · 2024-12-02
> > > > >
> > > > > Dear Authors,
> > > > >
> > > > > Thank you for your thoughtful response. The view I related to Reviewer KfFA's suggestion is just advice. While CIRQRS already demonstrates its potential as an evaluation metric, continuing to refine and validate its broader applicability could strengthen its role in advancing the CIR domain even further. I encourage you to keep pursuing this exciting direction. Your work is a valuable contribution to the field, and I look forward to seeing its future impact.
> > > > >
> > > > > Best Regards,
> > > > > Reviewer RFmP

---

> ### Author Response · Authors · 2024-11-18
> **Response to Reviewer RFmP - W2**
>
> **W2**
>
> > Since the goal is to introduce a new evaluation metric for CIR, a more detailed review of its effectiveness is needed. For example, testing CIRQRS on other models could help show its efficiency.
>
> **A2**
>
> Thank you for your constructive suggestions! To check the efficiency of our metric CIRQRS, we evaluated each of the four baseline models (CLIP4CIR, BLIP4CIR, CoVR-BLIP, and SPRC) on two representative datasets, FashionIQ and CIRR, using our metric CIRQRS.
>
> | FashionIQ | Dress CIRQRS@10 | Dress CIRQRS@50 | Shirt CIRQRS@10 | Shirt CIRQRS@50 | TopTee CIRQRS@10 | TopTee CIRQRS@50 | Avg. CIRQRS@10 | Avg. CIRQRS@50 | Average |
> |:----------:|-----------------|-----------------|-----------------|-----------------|------------------|------------------|----------------|----------------|---------|
> | CLIP4CIR | 67.97 | 66.35 | 68.61 | 66.51 | 67.62 | 65.88 | 68.07 | 66.25 | 67.16 |
> | BiBLIP4CIR | 71.03 | 68.86 | 69.90 | 67.51 | 70.16 | 67.64 | 70.36 | 68.00 | 69.18 |
> | CoVRBLIP | 72.07 | 69.78 | 71.41 | 68.76 | 71.38 | 68.64 | 71.62 | 69.06 | 70.34 |
> | SPRC | 72.26 | 69.99 | 71.89 | 69.27 | 71.76 | 69.07 | 71.97 | 69.44 | 70.71 |
>
>
> | CIRR | CIRQRS@1 | CIRQRS@5 | CIRQRS@10 | CIRQRS@50 | CIRQRS s@1 | CIRQRS s@2 | CIRQRS s@3 | Mean 5 + 1 | Mean |
> |:----------:|----------|----------|-----------|-----------|------------|------------|------------|------------|-------|
> | CLIP4CIR | 69.42 | 67.92 | 66.89 | 63.38 | 74.78 | 72.86 | 71.57 | 71.35 | 69.55 |
> | BiBLIP4CIR | 68.70 | 66.74 | 65.50 | 61.62 | 73.11 | 71.96 | 71.07 | 69.92 | 68.39 |
> | CoVRBLIP | 76.08 | 72.80 | 71.07 | 66.04 | 77.20 | 74.27 | 72.32 | 75.00 | 72.83 |
> | SPRC | 75.29 | 72.31 | 70.69 | 65.85 | 76.89 | 74.12 | 72.26 | 74.60 | 72.49 |
>
> We calculated the CIRQRS@k using mean CIRQRS of the retrieved set (Equation 7).
>
> Both tables show that CIRQRS@K decreases as K increases, indicating that all CIR methods retrieve more irrelevant images as K grows. In the FashionIQ dataset, SPRC achieved the best performance, following the same trend observed in Table 4 of our manuscript. However, in the CIRR dataset, the trends differ from those in Table 5, CLIP4CIR outperforms Bi-BLIP4CIR, and CoVRBLIP surpasses SPRC. This suggests that, although Bi-BLIP4CIR and SPRC achieved higher Recall@K, they are less aligned with human preferences than CLIP4CIR and CoVRBLIP.
>
> As our new metric CIRQRS reflects human preferences, we additionally experimented on the HS-FashionIQ dataset to explain this trend. Here, we used the CIR baselines’ similarity metric (e.g., cosine similarity) as a score to see the alignment between the human preference on the HS-FashionIQ dataset.
>
> | Methods | Preference Rate |
> |------------|-----------------|
> | CLIP4CIR | 0.6608 |
> | BiBLIP4CIR | 0.6700 |
> | CoVRBLIP | 0.7276 |
> | SPRC | 0.7339 |
> | __CIRQRS__ | __0.7524__ |
>
> The table shows that the four baselines follow the same order, perfectly reflecting the trend observed in the CIRQRS table for FashionIQ. This result also indicates that **CIR models with higher CIRQRS scores correspond to higher human preference rates**, highlighting the effectiveness of CIRQRS as an evaluation metric. However, we haven’t created the Human-scored CIRR dataset, making it challenging to explain why CLIP4CIR and CoVRBLIP perform better on the CIRR dataset when evaluated with CIRQRS. We plan to create a human-scored dataset for not only the FashionIQ dataset but also CIRR, aiming to further advance the CIR community. Notably, CIRQRS achieved the strongest alignment with human preferences among all baselines.

---

> > ### Comment · Reviewer_RFmP · 2024-11-26
> >
> > I suggest integrating these results into the relevant sections of the main manuscript. Placing them within the appropriate context will make the insights more prominent and further strengthen the narrative around the metric’s effectiveness.

---

> ### Author Response · Authors · 2024-11-18
> **Response to Reviewer RFmP - W3**
>
> **W3**
>
> > In Tables 2 and 3, the paper wants to demonstrate that CIRQRS is more effective than R@5, but the process by which these data were calculated should be clarified in greater detail. Since R@5 is an important point here, it would also help to show a comparison of R@5 results on the FashionIQ dataset as in Table 4 with other methods.
>
> **A3**
>
> We calculated both CIRQRS and Recall@5 for each set in the HS-FashionIQ dataset, which extracted the correlation between the human-given score with CIRQRS and Recall@5. Here, the set of retrieved images in the HS-FashionIQ dataset is retrieved from the randomly selected CIR model from CLIP4CIR, Bi-BLIP4CIR, CoVRBLIP, and SPRC. Further details of HS-FashionIQ can be found in Section 4, Data Collection Method part.
>
> We define the CIRQRS of a set as mean CIRQRS of the retrieved set (Equation 7). Recall@5 for the set is determined by whether it includes the target image, while human scores are based on the relevance score provided by humans after evaluating the set with the given query.
>
> We omitted the Recall@5 result comparison in our paper, because previous work typically doesn’t include them in the evaluation. However, we agree that with the given context of our HS-FashionIQ dataset, a comparison of Recall@5 is appropriate. In the table below, we show the comparison between CIRQRS and the 4 baseline model CIR that we used. The result shows that CIRQRS has a noticeable performance advantage over the rest of the models and strengthens our conclusion and contribution.
>
>
> | | Dress | Shirt | TopTee | Mean |
> |------------|:--------:|:--------:|:--------:|:--------:|
> | Methods | Recall@5 | Recall@5 | Recall@5 | Recall@5 |
> | CLIP4CIR | 11.75 | 24.58 | 20.75 | 19.03 |
> | BiBLIP4CIR | 25.88 | 27.87 | 31.51 | 28.42 |
> | CoVRBLIP | 35.99 | 40.48 | 42.89 | 39.79 |
> | SPRC | 35.65 | 42.74 | 44.88 | 41.09 |
> | __CIRQRS__ | __39.91__ | __46.66__ | __48.60__ | __45.06__ |

---

> ### Comment · Reviewer_RFmP · 2024-11-26
>
> Thank you for the additional experiments and detailed results.  Including evaluations on multiple baseline models and datasets strengthens the evidence for CIRQRS's alignment with human preferences. The provided tables and analysis are excellent supplements to the original manuscript, effectively demonstrating the effectiveness of the proposed CIRQRS metric.

---

### Official Review · Reviewer_kfFA · 2024-11-03

**Soundness:** 3
**Presentation:** 4
**Contribution:** 2
**Rating:** 6
**Confidence:** 4

**Summary:**

This paper is about CIRQRS (Composed Image Retrieval Query Relevance Score), a novel evaluation metric designed for Composed Image Retrieval (CIR) tasks, where users search for images based on a reference image and text query. The proposal is to use CIRQRS instead of the traditional Recall@k metric, which only checks if a target image is retrieved in the set of retrieved images. Since typically only one image is the target one, Recall@k can't evaluate if the other retrieved images may be still relevant to the composed query. CIRQRS, instead, assesses the relevance of each retrieved image to the query. The metric is trained using a reward model and a self-paced learning approach, where the negative set is dynamically adjusted based on relevance, to ensure it can rank images by query relevance accurately. The paper also introduces human-annotated data defining the HS-FashionIQ dataset, where CIRQRS is also validated and showing state-of-the-art performance, i.e. better correlation with human preferences over Recall@k. The method is also evaluated on the standard FashionIQ and CIRR datasets, demonstrating state-of-the-art performance.

**Strengths:**

- The paper is timely for the CIR community, I found the paper well structured and well written.

- CIRQRS addresses a core limitation in current CIR evaluation metrics by focusing on user-perceived relevance rather than simply on retrieving a target image. Traditional metrics like Recall@k fail to account for how well the retrieved set of images matches the user’s desired attributes, often leading to retrievals that technically meet the query but don’t align with user intent.

- Experiments on the FashionIQ and CIRR datasets show clear state of the art performance.

- The creation of the HS-FashionIQ dataset is a valuable contribution to the CIR field.

**Weaknesses:**

- The idea of proposing a metric based on a specific method to automatically evaluate images of the set is interesting, but it raises issues regarding alignment. Proposing CIRQRS as a primary CIR metric can be deceptive for future proposed methods, as is derived from a method that aims to align with user judgments but cannot perfectly capture user-defined relevance. This creates a risk that future CIR models, optimized based on CIRQRS, will only be partially aligned with actual user expectations. Specifically, this metric’s reliance on training data and model biases (in this case, the HS-FashionIQ dataset) introduces a limited view of “relevance” that might not generalize well across diverse retrieval contexts. Consequently, using CIRQRS as the evaluation standard could narrow the objective of future CIR models to a version of relevance that aligns more closely with CIRQRS’s training biases than with diverse user intent, specifically that of the ~2,7k valid queries. I understand the reason to evaluate the entire set instead of the single target image, but I do not see why this specific method should be better to be used as metric instead of any other existing state of the art CIR methods and check their correlation. Perhaps a better approach would be to treat CIRQRS as a methodological advancement—an effective way to improve the relevance evaluation capability of retrieval models—rather than promoting it as a general-purpose metric for CIR.

- The novelty of the self-paced learning strategy is limited to the application of the general strategy (like for instance in Jiang, Lu, et al. "Self-paced learning with diversity" Neurips 2014) to the method.

Typos
- page 6, FasionIQ should be FashionIQ.

**Questions:**

- I understand the reason to evaluate the entire set instead of the single target image, but I do not see why this specific method should be better to be used as metric instead of any other existing CIR methods. Perhaps a better approach would be to treat CIRQRS as a methodological advancement—an effective way to improve the relevance evaluation capability of retrieval models—rather than promoting it as a general-purpose metric for CIR. Can you compare CIRQRS with existing CIR methods as evaluation metrics, discussing their respective advantages and limitations?

- Are the HS-FashionIQ dataset and code of the method going to be released?

- Do you see potential mitigation strategies for these alignment concerns, such as methods to reduce bias in the training data or plans to validate CIRQRS across a more diverse range of retrieval contexts?

---

> ### Author Response · Authors · 2024-11-18
> **Response to Reviewer kfFA - W1**
>
> We deeply appreciate your insightful comments and efforts in reviewing our manuscript. We marked our revisions in the manuscript with “blue” fonts. Here, we respond to each of your comments.
>
> **W1-1**
>
> > The idea of proposing a metric based on a specific method to automatically evaluate images of the set is interesting, but it raises issues regarding alignment. Proposing CIRQRS as a primary CIR metric can be deceptive for future proposed methods, as is derived from a method that aims to align with user judgments but cannot perfectly capture user-defined relevance. This creates a risk that future CIR models, optimized based on CIRQRS, will only be partially aligned with actual user expectations. Specifically, this metric’s reliance on training data and model biases (in this case, the HS-FashionIQ dataset) introduces a limited view of “relevance” that might not generalize well across diverse retrieval contexts. Consequently, using CIRQRS as the evaluation standard could narrow the objective of future CIR models to a version of relevance that aligns more closely with CIRQRS’s training biases than with diverse user intent, specifically that of the ~2,7k valid queries.
>
> **A1-1**
>
> We aim to create a new metric that is more human-centered than the currently widely used Recall@K, as previous research suggests that user satisfaction correlates with the relevance of the retrieved set [1]. Thus, we derived the CIRQRS metric to represent the relevance of each image corresponding to each query.
>
> We created the HS-FashionIQ dataset as a validation tool to assess metrics in CIR (Recall@K, CIRQRS) to measure the alignment of human preferences using the __validation set__ of the FashionIQ dataset. In particular, we __did not__ train CIRQRS using HS-FashionIQ; instead, we trained it on just the training set of FashionIQ as in other previous work. This ensures that no data biases arise from HS-FashionIQ. Furthermore, our proposed framework for training CIRQRS is robust and can be adapted to other datasets, thus incorporating other datasets and retrieval context could lead to improved results. Moreover, To validate this approach, creating a human-scored dataset for other datasets (e.g. HS-CIRR) is needed and can serve as future work.
>
> [1] Azzah Al-Maskari and Mark Sanderson. A review of factors influencing user satisfaction in information retrieval. Journal of the American Society for Information Science and Technology, 61(5): 859–868, 2010
>
> **W1-2**
>
> > I understand the reason to evaluate the entire set instead of the single target image, but I do not see why this specific method should be better to be used as metric instead of any other existing state of the art CIR methods and check their correlation. Perhaps a better approach would be to treat CIRQRS as a methodological advancement—an effective way to improve the relevance evaluation capability of retrieval models—rather than promoting it as a general-purpose metric for CIR.
>
> **A1-2**
>
> Thank you for pointing this out. We agree that CIRQRS could be treated as a methodological advancement. However, we found that the existing widely used evaluation metric in CIR, Recall@K, does not align well with human preferences after creating the HS-Fashion IQ dataset. To address this, we designed CIRQRS to assign higher scores to more relevant images using our method. This work marks the first step in the CIR community toward proposing a new evaluation metric, which we believe will contribute to further advancements in the field.

---

> > ### Comment · Reviewer_kfFA · 2024-11-25
> > **W1 - reply**
> >
> > Thank you for your detailed responses to my review. Your clarifications and additional experiments have addressed several points effectively, though some concerns remain.
> >
> > While I appreciate your clarification that CIRQRS is not trained on HS-FashionIQ and your citation of research showing correlation between user satisfaction and retrieval set relevance, my core concern about alignment remains.
> >
> > Even if CIRQRS is trained only on FashionIQ's training set, the fundamental issue persists: future CIR models might optimize for CIRQRS scores rather than true user intent. This could create a feedback loop where models become increasingly aligned with CIRQRS's specific interpretation of relevance rather than broader user expectations. Your suggestion of creating additional human-scored datasets (for instance HS-CIRR) is valuable and could help mitigate this, but I believe this alignment challenge deserves more discussion also in the paper.

---

> ### Author Response · Authors · 2024-11-18
> **Response to Reviewer kfFA - W2**
>
> **W2**
>
> >The novelty of the self-paced learning strategy is limited to the application of the general strategy (like for instance in Jiang, Lu, et al. "Self-paced learning with diversity" Neurips 2014) to the method.
>
> **A2**
>
> We agree that previous self-paced learning shared a similar idea of selecting easy-to-hard samples. However, the key differences between CIRQRS and them are as follows.
>
> Firstly, our proposed training objective shows that representation learning can be done with preference optimization instead of the widely used[1][2] contrastive loss in CIR model training. This approach is more efficient, as each query only requires a single negative image instead of multiple negative images.
>
> However, it is not trivial to select proper negative images, as only one or a few target images are annotated in the CIR dataset. Also, random selection from the corpus is suboptimal, as seen in Table 7. To address this, we propose a method to appropriately select the negative images for each training step to boost and stabilize the training with ideas stemming from self-paced learning.
>
> Previous self-paced learning works [3][4][5] (1) use __multiple__ samples as negatives, and some of them are __pre-defined__, and (2) select easy-to-hard samples as just __top K__ samples based on their criteria (e.g., loss, distance). However, in CIR, negative images are not predefined, so it is challenging to avoid false-negative images (relevant but not a target) from being selected as a negative sample. To address it, we fixed the target image as positive, which aims for images similar to the target to receive higher/similar scores as training progresses. Consequently, we define the negative set as the __top K images with scores lower than the target image__ to ensure that relevant images are excluded from selection as negatives.
>
> We conducted experiments to support this idea by comparing (1) the original CIRQRS training, (2) INCLUDE, where the top $n_{neg}$ images are defined as a negative set, like existing self-paced learning works. The result graph is shown in Appendix B.1, Figure 6. Our simple yet effective new strategy for defining the negative set enhances the training. Defining just top $n_{neg}$ images as a negative set could include the relevant image. We additionally conducted a qualitative analysis on Appendix B.2 in our revised manuscript.
>
> Our approach also could be applied to other models and supervised information retrieval tasks, as it addresses the fundamental challenge of comparing (i.e., scoring) query and candidate embeddings. The ‘difficulty’ of images in this context depends on the current learned embedding space of the model, making our method adaptable across different models and retrieval tasks.
>
>
>
> [1] Bai Y. et. al. Sentence-level prompts benefit composed image retrieval. In The Twelfth International Conference on Learning Representations, 2024.
>
> [2] Liu Z. et. al. Bi-directional training for composed image retrieval via text prompt learning. In Proceedings of the IEEE/CVF Winter Conference on Applications of Computer Vision,
>
> [3]Jiang, Lu, et al. "Self-paced learning with diversity." Advances in neural information processing systems 27 (2014).
>
> [4] Supancic, James S., and Deva Ramanan. "Self-paced learning for long-term tracking." Proceedings of the IEEE conference on computer vision and pattern recognition. 2013.
>
> [5] Liu, Kangning, et al. "Multiple instance learning via iterative self-paced supervised contrastive learning." Proceedings of the IEEE/CVF Conference on Computer Vision and Pattern Recognition. 2023.

---

> > ### Comment · Reviewer_kfFA · 2024-11-25
> > **W2**
> >
> > The detailed explanation of the differences from previous self-paced learning approaches and the new experimental results in Appendix B.1 and B.2 effectively address my concern. Thank you.

---

> ### Author Response · Authors · 2024-11-18
> **Response to Reviewer kfFA - Q1**
>
> **Q1**
>
> >Can you compare CIRQRS with existing CIR methods as evaluation metrics, discussing their respective advantages and limitations?
>
> **A3**
>
> Thank you for your suggestion. We can compare CIRQRS with existing CIR methods as evaluation metrics if we treat their similarity scores as relevance scores. However, the key distinction with CIRQRS is that we train the model to assign higher scores to images relevant to the query—even if they are not the target image. In contrast, existing methods focus primarily on ranking the target image as high as possible.
>
> To evaluate this, we conducted an experiment comparing CIRQRS with other baselines on the HS-FashionIQ dataset. The results, shown in the table, indicate that CIRQRS has the strongest alignment with human preferences compared to the other baselines.
>
> | Methods | Preference Rate |
> |------------|-----------------|
> | Recall@5 | 0.5817 |
> | CLIP4CIR | 0.6608 |
> | BiBLIP4CIR | 0.6700 |
> | CoVRBLIP | 0.7276 |
> | SPRC | 0.7339 |
> | __CIRQRS__ | __0.7524__ |

---

> > ### Comment · Reviewer_kfFA · 2024-11-25
> > **Q1**
> >
> > The provided results demonstrate the approach performance effectively. Thank you.

---

> ### Author Response · Authors · 2024-11-18
> **Response to Reviewer kfFA - Q2, Q3**
>
> **Q2**
>
> >Are the HS-FashionIQ dataset and code of the method going to be released?
>
> **A4**
>
> We plan to release the HS-FashionIQ dataset, our full training code, and model weights of our experiment upon acceptance.
>
>
> **Q3**
>
> >Do you see potential mitigation strategies for these alignment concerns, such as methods to reduce bias in the training data or plans to validate CIRQRS across a more diverse range of retrieval contexts?
>
> **A5**
>
> We __did not__ train CIRQRS on the HS-FashionIQ dataset, thus eliminating the risk of an alignment concern. Accordingly, following your suggestions, to evaluate the CIRQRS across a more diverse range of retrieval contexts, we conducted two experiments to assess CIRQRS’s zero-shot capabilities.
>
> (1) We trained the CIRQRS with the CIRR dataset and tested it on the CIRCO dataset. The results show that CIRQRS outperforms two zero-shot CIR methods, highlighting its generalizability.
>
>
> | | mAP@k=5 | mAP@k=10 | mAP@k=25 | mAP@k=50 |
> |--------|---------|----------|----------|----------|
> | SEARLE[1] | 11.68 | 12.73 | 14.33 | 15.12 |
> | CIReVL[2] | 18.57 | 19.01 | 20.89 | 21.80 |
> | __CIRQRS__ | __20.85__ | __21.48__ | __23.35__ | __24.31__ |
>
> [1] Baldrati, Alberto, et al. "Zero-shot composed image retrieval with textual inversion." Proceedings of the IEEE/CVF International Conference on Computer Vision. 2023.
>
> [2] Karthik, Shyamgopal, et al. "Vision-by-language for training-free compositional image retrieval." International Conference on Learning Representations (ICLR), 2024.
>
> (2) Additionally, we compared CIRQRS with CIR baselines on the HS-FashionIQ dataset to evaluate alignment with human preferences. The results indicate that CIRQRS trained with the FashionIQ dataset achieves the best performance among all other methods. Interestingly, even though the CIRQRS is trained on the CIRR dataset, it shows stronger alignment with human preferences than other CIR baselines that are trained on the FashionIQ dataset, which has a data distribution that is more similar to that of the HS-FashionIQ dataset than the CIRR dataset. These findings demonstrate that CIRQRS is not only less biased by training data but also exhibits strong generalizability.
>
>
> | Methods | Preference Rate |
> |-------------------|-----------------|
> | CLIP4CIR | 0.6608 |
> | BiBLIP4CIR | 0.6700 |
> | CoVRBLIP | 0.7276 |
> | SPRC | 0.7339 |
> | __CIRQRS(CIRR)__ | __0.7377__ |
> | __CIRQRS(FashionIQ)__ | __0.7524__ |

---

> > ### Comment · Reviewer_kfFA · 2024-11-25
> > **Q2, Q3**
> >
> > I agree, these additional experiments are particularly interesting indicating a strong method generalisability, although they do not address completely the fundamental problem of alignment.

---

> > > ### Author Response · Authors · 2024-11-26
> > > **About alignment concern**
> > >
> > > Thank you for your insightful comments and clarification. We completely agree that discussing the alignment you mentioned is crucial and are happy to discuss this important point in the paper.
> > >
> > > First, if human-scored datasets are available for all CIR datasets, designing a new evaluation metric in this field would be unnecessary. For example, as shown in experiments with CIR baselines on the HS-FashionIQ dataset (the table in Q1-A3), the human preference rate for different CIR models can be calculated, enabling direct evaluation based on human preferences. However, as the number of datasets grows, creating such human-scored datasets consistently becomes increasingly impractical.
> > >
> > > To address this, we designed training objectives to assign higher scores to more relevant images. Given the limitations of existing CIR datasets—where only one or a few target images are annotated per query—we treated the target as positive and defined a negative set by selecting images that are relatively less relevant than the target. This approach ensures that the training aligns with relevance-based evaluation.
> > >
> > > Our experiments demonstrated that CIRQRS achieves the best alignment with human preferences on the FashionIQ dataset, as shown by a human preference rate of 0.7524 on the HS-FashionIQ dataset. This result directly validates CIRQRS for the FashionIQ dataset, and we believe it would also align well with other datasets. However, we acknowledge the need to create additional human-scored datasets to validate this claim thoroughly.
> > >
> > > We also recognize that a metric perfectly aligned with human preferences should achieve a preference rate closer to 1. If such a metric exists, the alignment concern you raised would be fully addressed. To mitigate the current limitations of CIRQRS, we suggest combining it with other metrics, such as Recall@k, to evaluate CIR models holistically. This approach ensures that CIR models are assessed not only on their ability to retrieve the target image but also on their capacity to rank relevant images higher, avoiding potential biases inherent only to CIRQRS.
> > >
> > > As the first evaluation metric in the CIR field to explicitly consider human preferences, we believe CIRQRS represents an important initial step toward advancing the field.
> > >
> > > In summary, we plan to incorporate the following points in the discussion section of our paper regarding alignment:
> > > 1. Propose the creation of additional human-scored datasets for thorough evaluation.
> > > 2. Suggest using both CIRQRS and Recall@k to address the limitations of each metric and provide a more comprehensive evaluation
> > >
> > > We appreciate the opportunity to address your concerns and kindly ask you to consider revising your rating if you feel they have been resolved. We are happy to continue the discussion if you have additional questions. Thank you.

---

> > > > ### Author Response · Authors · 2024-12-02
> > > >
> > > > Dear Reviewer,
> > > >
> > > > Thank you once again for your detailed comments and valuable feedback on our work. We would greatly appreciate it if you could let us know whether our responses have sufficiently addressed your concerns and if you have any additional comments on the paper.
> > > >
> > > > Best regards,
> > > > Authors.

---

> > > > > ### Comment · Reviewer_kfFA · 2024-12-03
> > > > > **Thank you**
> > > > >
> > > > > Thank you for the effort and the detailed rebuttal. I raised my score from 5 to 6 following the authors' clarifications.

---

### Author Response · Authors · 2024-11-18
**General Response**

Dear reviewers and AC,

We sincerely appreciate your valuable time and effort spent reviewing our manuscript.

As the reviewers highlighted, we believe our paper is well-structured (kfFA, RFmP), and presents a novel evaluation metric, timely in CIR field (kfFA), and also shows strong empirical results on representative datasets (all reviewers). In particular, we see our work as a meaningful first step in the CIR community toward establishing a new evaluation metric, which we believe will contribute to further advancements in the field.

We have carefully revised the manuscript according to your suggestions, with all changes highlighted in blue. The major changes are listed below.

1. We added additional sentences in the Method section to emphasize the novelty of our work.
2. We included both quantitative and qualitative experimental results in Appendix B to support our method component.
3. We renamed CIRQRS to CIRQRS-Model when used as a model (in Section 5.3), to avoid confusion, as CIRQRS is originally designed as the metric.

In addition to submitting a revised paper, we reply to each reviewer individually to address their questions and concerns. We hope that these responses together with the revised manuscript clear up any confusion and resolve all issues that the reviewers had, and that you will consider increasing your rating for our paper.

---

> ### Author Response · Authors · 2024-11-23
> **General Response**
>
> Dear reviewers and AC,
>
> Thank you for your valuable time and effort in reviewing our manuscript.
>
> As noted by the reviewers (RFmP and 1H4V) and acknowledged by us, it is crucial to evaluate CIR models using our proposed metric, CIRQRS. We have added Section 5.4, **Evaluation of CIR Models with CIRQRS**. This section presents the evaluation results of existing CIR models using CIRQRS and demonstrates that **CIR models with higher CIRQRS scores correspond to higher human preference rates.** These findings underscore the effectiveness of CIRQRS as an evaluation metric.
>
> The corresponding revisions have been highlighted in blue in the updated manuscript.
>
> We believe these changes have substantially enhanced the clarity and technical depth of the paper. If you have any additional questions or concerns, please feel free to reach out.

---

### Meta-Review · Area_Chair_tsvR · 2024-12-20

**Metareview:**

The article introduces the Composed Image Retrieval Query Relevance Score (CIRQRS), a new evaluation metric for Composed Image Retrieval (CIR) tasks. The authors start from the main deficiency of Recall@k, which only checks if a target image is retrieved. CIRQRS instead evaluates the relevance of each retrieved image to the query. It is a metric trained  using a reward model and self-paced learning to adjust the negative image set based on relevance. The paper also proposes a new human-scored Fashion Image Quality dataset (HS-FashionIQ) and validates CIRQRS on it.

The submitted manuscript has a number of strong points recognized by the reviewers. They note that the paper is well-structured and effectively written, making it accessible and its contributions clear. Reviewers also find interesting the approach to addressing the main limitation of transitional metrics like Recall@k and proposing to concentrate on user-perceived relevance. Finally, reviewers appreciate the value of the proposed HS-FashionIQ dataset and note the excellent performance of CIRQRS on it and CIRR.


The paper however has the following weak points that outweigh its strengths:
+ **Alignment Issues**: The contribution of CIRQRS hinges on user judgments represented in the training data, which poses risks of misalignment with actual user expectations of future CIR models with as they may optimize based on a narrow view of relevance shaped by CIRQRS’s training biases.

+ **Novelty and Clarity**: Multiple reviewers express concerns regarding the novelty, both specifically that of the the self-paced learning strategy and in a general sense that the novelty of the overall contribution is not effectively communicated in the original submission. There is inconsistency in the terminology regarding CIRQRS, being referred to interchangeably as a score, metric, and model, which impedes understanding. In rebuttal the authors revised the manuscript to address some of these concerns, however the overall consensus is that the manuscript is in need of additional revision in order to better communicate the novel aspects of the work.

+ **Correlation with Human Scores**: Correlating CIRQRS with human preferences appears inadequate, which risks misleading interpretations of effectiveness of the proposed approach. In particular, demonstrating that the CIRQRS metric is more correlated with human preference when compared to Recall@k is not sufficient to broadly validate the value of a new, trained metric like CIRQRS.

Overall, reviewers recognize that the paper has a number of interesting ideas in it. However, the general consensus is that the work is in need of significant revision in terms of clarity and methodology before it should be considered for acceptance.

**Additional Comments On Reviewer Discussion:**

There was some back-and-forth between reviewers and authors during the discussion period. The general consensus is that the work needs to better articulate novelty and improve clarity in the technical descriptions of the proposed methodology. While most reviewers recognized the positive aspects of the contribution, none of them had strong feelings towards acceptance.

---

### Decision · Program_Chairs · 2025-01-22

Reject